# Clinicopathologic and molecular predictors of survival in *BRCA*-deficient tubo-ovarian high-grade serous carcinoma

*BRCA*-associated homologous recombination deficiency (HRD) is present in ~50% of high-grade serous carcinomas (HGSC) and predicts sensitivity to platinum-based therapy. However, there is little understanding of why some patients with *BRCA*-deficient tumors experience poor outcomes. In a large HGSC cohort (n = 1389) including 282 individuals with pathogenic germline *BRCA* variants (g*BRCA*pv), residual disease after primary surgery has limited prognostic effect in g*BRCA*pv-carriers compared to non-carriers, and prognostic outcomes differ based on the mutation location within functional domains of the *BRCA* genes. Multi-omic profiling is performed on 154 tumors, enriched for patients with *BRCA*-deficient tumors that experienced short overall survival (≤ 3 years, n = 42). Patients with *BRCA2*-deficient HGSC and loss of *NF1* survive twice as long as those without *NF1* loss, whereas *PIK3CA*, *RAD21* and *MYC* amplification define *BRCA2*-deficient HGSC with exceptionally short survival. Patients with *BRCA1*-deficient HGSC and a more elevated HRD score survive significantly longer. *BRCA1*-deficient tumors in short survivors have evidence of immunosuppressive c-kit signaling and EMT. Our findings confirm that outcome is not determined by *BRCA* status alone, but rather a combination of co-occurring genomic alterations, the extent of DNA repair deficiency, and the tumor-immune microenvironment.

The identification of clinical and molecular determinants of survival in patients with cancer has the dual benefit of finding biomarkers that may guide patient management or provide novel therapeutic opportunities. Until relatively recently, the identification of prognostic biomarkers in ovarian cancer has been confounded by a lack of appreciation of the distinctly different molecular characteristics of the various histologic subtypes that make up epithelial ovarian cancer[1]. Evaluating histologically homogenous sets of ovarian tumors has been critical in deciphering the prognostic importance of proteins such as p53[2,3] and WT1[4], and identifying genetic risk loci[5–12].

High-grade serous carcinoma (HGSC) is the most common histotype, accounting for approximately 70% of ovarian cancer deaths in Western countries[13–16]. Homologous recombination-mediated DNA repair deficiency (HRD) is frequent in HGSC and is most often associated with mutations in *BRCA1* and *BRCA2*[17–19]. Approximately fifty percent of HGSC are regarded to have HRD, a feature that can be inferred through specific patterns of genomic scarring in tumor cells[13,20–25]. HRD leads to genomic instability and tumorigenesis, providing a vulnerability in tumor cells with increased sensitivity to double-strand DNA breaks that can be exploited therapeutically[26–28]. As a result, platinum-based chemotherapy and poly (ADP-ribose) polymerase inhibitor (PARPi) maintenance therapy are generally more effective in patients with HRD tumors[28–34].

While HRD status is informative, accurate prediction of treatment response and survival in HGSC cannot be simply determined by the presence or absence of mutations in genes associated with HR DNA repair[26,35]. The initial survival advantage for carriers of pathogenic germline *BRCA1* variants (g*BRCA1*pv) diminishes over time, with fewer

✉e-mail: tibor.zwimpfer@petermac.org; Dale.Garsed@petermac.org

g*BRCA1*pv-carriers surviving 10 years after diagnosis than either g*BRCA2*pv-carriers or non-carriers[33,36,37]. Factors associated with survival outcome in HGSC include residual disease following cytoreductive surgery[16,38–40], the molecular subtype of the tumor[41], age at diagnosis[42], and the extent of T- and B-cell infiltration into tumors[43,44]. In germline pathogenic variant carriers, the location of mutations within *BRCA1* or *BRCA2,* or the retention of the wildtype allele in the tumor can result in a hypomorphic phenotype associated with resistance to platinum-based therapy[45–49]. Furthermore, revertant mutations restoring *BRCA1* and *BRCA2* function contribute to acquired resistance to platinum-based therapy and PARPis, impacting treatment response and patient outcomes[50–52].

Comparing patients who represent the extremes of survival outcomes may provide increased sensitivity to identify prognostic biomarkers that are relevant to a wider patient population[53]. Using this approach, we have recently shown that plasma cell infiltration and other molecular changes, including co-loss of *BRCA* and the tumor suppressor *RB1*, are associated with especially long-term survival in HGSC[22,54,55].

The current study evaluates *BRCA*-deficient HGSC by first focusing on g*BRCA*pv-carriers and then expanding to include somatic mutations and promoter methylation in *BRCA1/2*, and other key HR genes, as well as evaluating tumor HRD status. We focus on patients with either poor or favorable survival outcomes, harnessing the value of analyzing patients with exceptional survival outcomes while comparing cohorts that are as similar as possible in other respects.

## Results

### Association of residual disease with prognosis is attenuated in g*BRCA*pv-carriers

Pathogenic germline *BRCA* variants (g*BRCA*pv) were identified in 20% of patients in the Australian Ovarian Cancer Study (AOCS) cohort ($n = 282/1389$) (Table 1, Supplementary Data 1 and 2). In applying a survival model, there was evidence that the proportional hazards assumption did not hold ($P < 0.001$), thus an Accelerated Failure Time (AFT) model[56] was used with results reported as Time Ratios (TR; see Methods), where TR > 1 indicates longer time to progression or death, and a TR < 1 indicates shorter survival or time to progression. Patients with g*BRCA*pvs exhibited improved overall survival (OS; TR: 1.53, 95% CI: 1.33–1.76, $P < 0.001$) and progression-free survival (PFS; TR: 1.34 95% CI: 1.28–1.53, $P < 0.001$) compared with non-carriers (Supplementary Tables 1 and 2).

We considered whether clinical characteristics differed by germline *BRCA* status and found a statistically significant interaction with residual disease status (P-interaction = 0.011; Supplementary Table 3). Using this interaction term, we found that the negative effect of residual disease after cytoreductive surgery on OS was less pronounced in g*BRCA*pv-carriers (TR: 0.87, 95% CI: 0.72–1.06, $P = 0.162$) than in non-carriers (TR: 0.51, 95% CI: 0.44–0.59, $P < 0.001$; Fig. 1A, Table 2). The importance of residual disease for survival in non-carriers was confirmed in the independent OTTA cohort ($n = 1004$, g*BRCA*pv-carriers = 221, 22%; Fig. 1B, Supplementary Figs. 1 and 2a). A subanalysis excluding patients who received PARPi treatment in the first-line setting showed consistent results (Supplementary Table 3).

Because the violation of proportional hazards suggested time-dependency, we examined the shape of the survival curves. Both OS curves (Fig. 1A, B) showed that the steepest decline among g*BRCA*pv-carriers with no residual disease was between years 2–3. Examination of the PFS curve (only available for the AOCS cohort, Supplementary Fig. 2b), similarly demonstrated early separation between g*BRCA*pv-carriers and non-carriers that began to narrow after approximately two years.

We next assessed the relationship of residual disease and *BRCA* status to known immune and molecular features associated with survival, including tumor-infiltrating lymphocytes (TIL)[44,57], *RB1* loss[22,54,58],

and transcriptional molecular subtypes[41]. Non-carriers with residual disease had an inverse association with high CD8 + TIL density ($P = 0.016$), with 38.3% of tumors classified as having low or no TIL (Supplementary Fig. 3, Supplementary Table 4). This group also showed an inverse association with the C4/differentiated (C4.DIF) molecular subtype ($P = 0.010$; Supplementary Fig. 3). We observed an association between the C1/mesenchymal (C1.MES) molecular subtype and residual disease as previously reported[59], but this was only statistically significant among non-carriers ($P = 0.005$). *RB1* loss was associated with g*BRCA*pv-carriers without residual disease ($P < 0.001$; Supplementary Fig. 3).

Although no statistically significant interaction between neoadjuvant chemotherapy (NACT) and *BRCA* status was observed (P-interaction=0.12; Supplementary Table 1), there was evidence of heterogeneity of effect in these subgroups. Among participants who did not receive NACT, g*BRCA*pv-carriers showed a survival benefit compared to non-carriers (TR: 1.60, 95% CI: 1.37–1.87, $P < 0.001$; Supplementary Table 5, Supplementary Fig. 4). In contrast, the OS benefit in g*BRCA*pv-carriers versus non-carriers was not statistically significant in the NACT group (TR: 1.39 and 1.17, 95% CI: 0.75–2.60 and 0.62–2.21, $P = 0.298$ and $P = 0.634$ respectively, compared to non-carriers who did not receive NACT).

### g*BRCA*pv location and type are associated with survival and therapy response

Mutations located in various functional domains of *BRCA1* and *BRCA2* have been associated with differences in survival and responses to PARPi in ovarian cancer[45,46,60]. The mutation type and location of g*BRCA*pvs was ascertained for 240 of the patients in the AOCS cohort from their clinical records and/or previous sequencing analyses[22,58,61,62] (Supplementary Fig. 5a, b and Supplementary Data 2). Following adjustment for FIGO stage, residual disease status, primary site, age, and first-line treatment, patients with g*BRCA1*pvs in exon 10 had a statistically significant improved OS and PFS (TR: 1.54 and 1.49, 95% CI: 1.19–2.00 and 1.16–1.91, $P < 0.001$ and $P = 0.002$, respectively), but the association was attenuated for those with variants outside exon 10 (TR: 1.21 and 1.18, 95% CI: 0.97–1.51 and 0.96–1.46, $P = 0.09$ and $P = 0.12$, respectively) compared to non-carriers (Table 3). More specifically, pathogenic variants in the DNA binding domain (DBD) of *BRCA1*, located in exon 10, were associated with an OS and PFS benefit compared to non-carriers (TR: 1.60 and 1.58, 95% CI: 1.14–2.25 and 1.15–2.18, $P = 0.005$ and $P = 0.006$, respectively; Table 3). In contrast, the OS and PFS benefit was not statistically significant for patients with pathogenic variants in the Really Interesting New Gene (RING) (TR: 1.28 and 1.15, 95% CI: 0.87–1.90 and 0.82–1.61, $P = 0.216$ and $P = 0.419$, respectively) and C-terminal domains of *BRCA1* (BRCT) (TR: 1.35 and 1.43, 95% CI: 0.83–2.20 and 0.90-2.26, $P = 0.222$ and $P = 0.126$, respectively), located outside of exon 10. Notably, founder variants were enriched in the *BRCA1* RING domain (59.3%, $P < 0.0001$), whereas no such enrichment was observed for *BRCA2* domains (Supplementary Fig. 5c, d and Supplementary Data 2).

Patients with *BRCA1* variants in exon 10 have been reported to have poorer outcomes[48] due to expression of an alternative splice isoform called *BRCA1*-delta11q (Δ11q) that bypasses almost all of exon 10 of *BRCA1* (historically referred to as exon 11). To explore this further, we assessed *BRCA1* isoform expression in our multi-omics cohort ($n = 154$) using the bulk RNA sequencing reads spanning the exon 10 to exon 11 junction (Fig. 2A, Supplementary Data 3 and 4, Supplementary Notes). The Δ11q isoform was widely expressed regardless of *BRCA*-status, but patients with *BRCA1* variants in exon 10 had significantly higher proportions of Δ11q transcripts relative to canonical transcripts ($P = 0.011$; Fig. 2B). Patients with *BRCA1* variants in exon 10 were classified as having high ($n = 10$) or low ($n = 9$) *BRCA1* Δ11q expression following a median split. Patients with high Δ11q expression had a shorter survival (median OS 2.74 years) compared to those with low

## Table 1 | Baseline characteristics of the clinicopathological features from patients with HGSC of the Australian Ovarian Cancer Study (AOCS) cohort

| Characteristics | n = 1389 n (%) |
| --- | --- |
| **Age at diagnosis (years)** | |
| Median | 61 |
| Range | 24–87 |
| Unknown | 7 (0.5) |
| **Germline *BRCA* status** | |
| Wildtype | 1107 (79.7) |
| gBRCA1pv | 175 (12.6) |
| gBRCA2pv | 107 (7.7) |
| **Grade** | |
| G3 | 1100 (79.2) |
| G2 | 237 (17.1) |
| Unknown | 52 (3.7) |
| **FIGO stage** | |
| III–IV | 1193 (85.9) |
| I–II | 134 (9.6) |
| Unknown | 62 (4.5) |
| **Primary site** | |
| Ovary | 1008 (72.6) |
| Peritoneum | 215 (15.5) |
| Fallopian tube | 140 (10.1) |
| Unknown | 26 (1.9) |
| **Surgery** | |
| Primary cytoreductive surgery | 991 (71.3) |
| Interval cytoreductive surgery | 299 (21.5) |
| Other | 70 (5) |
| Unknown | 29 (2.1) |
| **Residual disease status** | |
| Residual disease | 829 (59.7) |
| No residual disease | 467 (33.6) |
| Unknown | 93 (6.7) |
| **Neoadjuvant chemotherapy** | |
| No | 1060 (76.3) |
| Yes | 322 (23.2) |
| Unknown | 7 (0.5) |
| **PARP inhibitor 1st line** | |
| No | 1350 (97.2) |
| Yes | 39 (2.8) |
| **Progression-free survival (months)** | |
| Median | 15 |
| Range | 0–285 |
| Unknown | 11 (0.8) |
| **Overall survival (months)** | |
| Median | 38 |
| Range | 1–290 |
| Unknown | 11 (0.8) |
| **Status** | |
| Deceased | 984 (70.8) |
| Alive | 393 (28.3) |
| Unknown | 12 (0.9) |

*G2* Grade 2, *G3* Grade 3, *OS* Overall survival, *gBRCApv* pathogenic germline BRCA variant.
Source data are provided as a Source Data file.

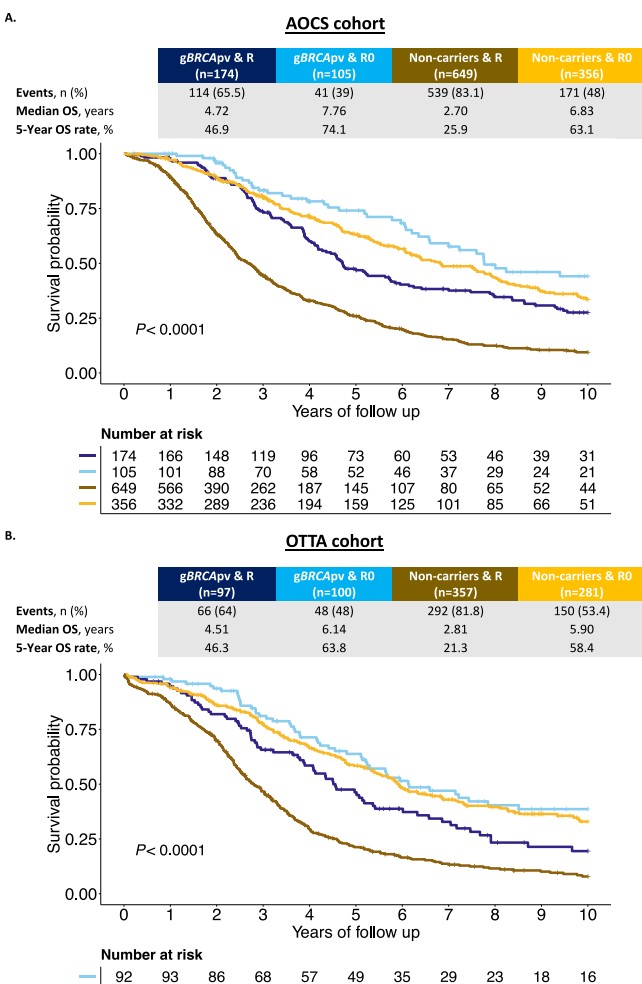

**Fig. 1 | *BRCA* status and residual disease as predictors of overall survival in HGSC (AOCS and OTTA cohort).** Kaplan–Meier survival curve for the interaction term *BRCA* and Residual status from patients of **A** the Australian Ovarian Cancer Study (AOCS) cohort (*n* = 1284 patients) and **B** the Ovarian Tumor Tissue Consortium (OTTA) cohort (*n* = 835 patients). *P* values calculated by log-rank test. Source data are provided as a Source Data file. R=Residual disease, R0 = No residual disease, gBRCApv=pathogenic germline BRCA variant, n=Number of patients, OS=Overall survival.

Δ11q expression (median OS not reached), although this was not statistically significant (*P* = 0.083) and was not associated with differences in the HRD sum score (Fig. 2C, D and Supplementary Data 5).

Overall, patients with gBRCA2pv had an improved OS compared to non-carriers, regardless of mutation location (Table 3). The only exception was the small group (*n* = 13) with pathogenic variants in the DNA binding domain (DBD) of *BRCA2*, located outside of exon 11, who did not show a statistically significant OS or PFS benefit compared to non-carriers (TR: 0.79 and 0.81, 95% CI: 0.39–1.63 and 0.43–1.51, *P* = 0.528 and *P* = 0.506, respectively).

The type of mutation in *BRCA1* and *BRCA2* also plays a predictive role in response to PARPi therapy in ovarian cancer[45,60]. In our analysis, pathogenic variants in *BRCA1* exon 10 and *BRCA2* exon 11 were more likely to be truncating (98.6% and 92.3%) than those outside these exons (60% and 76.3%, *P* < 0.001 and *P* = 0.032 respectively; Fig. 2E, F). *BRCA1* and *BRCA2* domains associated with prolonged survival were more likely to have truncating variants than missense or splice site variants (*P* < 0.001 and *P* = 0.067, respectively; Supplementary Fig. 5e, f). Consistent with these findings, analysis of mutation type

**Table 2 | Multivariable Accelerated Failure Time (AFT) model of *BRCA* and residual disease status and clinicopathological predictive features on overall survival in patients (n = 1377) from the Australian Ovarian Cancer Study (AOCS) cohort**

| Feature | Factor | Number | Univariable | | | | Multivariable | | | |
|---|---|---|---|---|---|---|---|---|---|---|
| | | | TR | 95%CI | | *P*-value | TR | 95%CI | | *P*-value |
| | | | | lower | upper | | | lower | upper | |
| g*BRCA*pv & Residual status | Non carriers & R0 | 356 | - | - | - | - | - | - | - | - |
| | Non carriers & R | 649 | 0.42 | 0.37 | 0.48 | <0.001 | 0.51 | 0.44 | 0.59 | <0.001 |
| | g*BRCA*pv carriers & R0 | 105 | 1.31 | 1.03 | 1.66 | 0.028 | 1.18 | 0.92 | 1.50 | 0.191 |
| | g*BRCA*pv carriers & R | 174 | 0.82 | 0.68 | 0.98 | 0.028 | 0.87 | 0.72 | 1.06 | 0.162 |
| FIGO stage | I + II | 134 | - | - | - | - | - | - | - | - |
| | III + IV | 1183 | 0.34 | 0.28 | 0.42 | <0.001 | 0.59 | 0.47 | 0.74 | <0.001 |
| Primary site | Ovary | 1002 | - | - | - | - | - | - | - | - |
| | FT | 135 | 1.28 | 1.04 | 1.58 | 0.023 | 1.10 | 0.89 | 1.36 | 0.364 |
| | Peritoneum | 215 | 0.66 | 0.57 | 0.77 | <0.001 | 0.82 | 0.71 | 0.94 | 0.005 |
| Age at diagnosis | Years Spline 1 | 1370 | 0.99 | 0.98 | 1.00 | 0.069 | 1.00 | 0.99 | 1.01 | 0.669 |
| | Years Spline 2 | | 0.99 | 0.97 | 1.00 | 0.142 | 0.98 | 0.97 | 1.00 | 0.029 |
| Surgery | Primary CS | 980 | - | - | - | - | - | - | - | - |
| | Interval CS | 299 | 0.83 | 0.72 | 0.95 | 0.007 | 0.89 | 0.48 | 1.64 | 0.703 |
| | Other | 69 | 1.27 | 0.99 | 1.63 | 0.065 | 1.13 | 0.84 | 1.53 | 0.418 |
| Neoadjuvant CHT | No | 1048 | - | - | - | - | - | - | - | - |
| | Yes | 322 | 0.83 | 0.72 | 0.95 | 0.006 | 0.96 | 0.53 | 1.75 | 0.897 |
| Grade | G2 | 237 | - | - | - | - | - | - | - | - |
| | G3 | 1088 | 1.22 | 1.05 | 1.41 | 0.009 | 1.07 | 0.93 | 1.22 | 0.347 |
| PARP inhibitor 1st line | No | 1338 | - | - | - | - | - | - | - | - |
| | Yes | 39 | 1.40 | 0.90 | 2.12 | 0.119 | 1.25 | 0.81 | 1.92 | 0.321 |

The model was fitted using a log-logistic distribution. Results are expressed as Time Ratios (TR) with corresponding 95% confidence intervals (CI). Two-sided *P*-values were derived from Wald tests. A TR > 1 indicates a longer survival time, whereas a TR < 1 indicates a shorter survival time. Age at diagnosis was modeled using restricted cubic splines with 3 knots and is presented as two spline terms. Source data are provided as a Source Data file.
*R* Residual disease, *R0* No residual disease, *G2* Grade 2, *G3* Grade 3, *OS* Overall survival, *gBRCApv* pathogenic germline BRCA variant, *TR* Time ratio, *CI* confidence interval, *CHT* chemotherapy, *CS* cytoreductive surgery, *FT* fallopian tube.

demonstrated that truncating g*BRCA*pv were associated with longer PFS and OS compared with non-carriers (PFS TR:1.48, 95% CI:1.27–1.73, *P* < 0.001; OS TR:1.58, 95%CI: 1.35–1.86, *P* < 0.001; Table 3), whereas missense, splice site, or structural variants did not show a significant survival advantage. Additionally, founder mutation status showed no association with mutation type in either *BRCA1* or *BRCA2* (Supplementary Fig. 5g, h and Supplementary Data 2).

### *NF1* gene alterations are associated with improved survival in *BRCA2*-deficient HGSC

To identify genomic features associated with short survival in HRD tumors, we compared tumor genomes and transcriptomes between short (OS ≤ 3 years, STS) and long-term (OS > 3 years, LTS) survival groups (Fig. 3A). Tumor genomes were classified as either *BRCA1*-deficient, *BRCA2*-deficient or *BRCA*-proficient, which incorporated germline and somatic alterations in *BRCA1* and *BRCA2*, as well as other well-defined HR genes, and tumor HRD status as determined by a mutational signature-based classifier (CHORD, Classifier of HOmologous Recombination Deficiency)[63] (Supplementary Notes and Supplementary Data 6–8). Consistent with previous reports[22,61,64,65], *CCNE1* amplifications (gene level copy number ≥7) were associated with *BRCA*-proficiency, and particularly the short-survival *BRCA*-proficient group (50%, $P_{adj}$ < 0.001; Fig. 3B, Supplementary Table 6). Similarly, as previously described[22,61], *BRCA*-proficient tumors had less genomic scarring and were associated with an older age at diagnosis compared to *BRCA1*-deficient and *BRCA2*-deficient tumors (Supplementary Fig. 6a, b). Gene methylation has been identified as a prognostic factor

in HGSC[66], but no significantly differentially methylated genes with corresponding up- or down-regulated gene expression were observed between STS and LTS groups in *BRCA1*- and *BRCA2*-deficient tumors (Supplementary Table 7 and Supplementary Notes).

Alterations in *NF1* were most common in *BRCA*-deficient tumors, regardless of survival group (*BRCA1* STS 43.8%, *BRCA1* LTS 33.3%, *BRCA2* STS 30%, *BRCA2* LTS 37.5%, *BRCA*-P STS 21.4%, *BRCA*-P LTS 14.3%, $P_{adj}$ = 0.061; Fig. 3C and Supplementary Table 6, Supplementary Data 9). Notably, gene breakage caused by large-scale deletions was enriched in *BRCA2*-deficient tumors in the LTS group. We hypothesized that not all alteration types equivalently disrupt gene function. Indeed, only 54.2% (26/48) of *NF1* alterations showed a locus-specific loss of heterozygosity (LOH) suggesting a loss-of-function (Supplementary Data 9 and Supplementary Notes). Concordantly, *NF1* mRNA expression varied in tumors according to the type of *NF1* alteration and was particularly depleted in those with locus-specific LOH (*P* < 0.0001; Supplementary Fig. 7a). Patients with tumors that harbored loss-of-function *NF1* alterations showed an improved survival compared to non-loss-of-function *NF1* alterations (median OS 11.92 years vs 3.84 years, *P* = 0.032; Supplementary Fig. 7b). In particular, the combination of both *BRCA2*-deficiency and loss-of-function *NF1* alteration (*n* = 11) was associated with the best survival outcome (median OS 16.96 years), almost twice as long as those with *BRCA2*-deficient tumors with no loss-of-function *NF1* alteration (median OS 8.84 years; Supplementary Fig. 7c and Supplementary Data 6).

NF1 protein expression was assessed by IHC in a larger cohort enriched for long-term survivors (*n* = 658; Supplementary Fig. 1). NF1

**Table 3 | Multivariable Accelerated Failure Time (AFT) model analysis of germline *BRCA* pathogenic variant (g*BRCA*pv) location and type with progression-free survival and overall survival in patients (n = 1377) from the Australian Ovarian Cancer Study (AOCS) cohort**

| Feature | Factor | Number | Progression-free survival | | | | Overall survival | | | |
|---|---|---|---|---|---|---|---|---|---|---|
| | | | | 95%CI | | | | 95%CI | | |
| | | | TR | lower | upper | *P*-value | TR | lower | upper | *P*-value |
| gBRCApv exon | Non carriers | 1096 | - | - | - | - | - | - | - | - |
| | gBRCA1pv Exon 10 | 68 | 1.49 | 1.16 | 1.91 | 0.002 | 1.54 | 1.19 | 2.00 | <0.001 |
| | gBRCA1pv outside Exon 10 | 81 | 1.18 | 0.96 | 1.46 | 0.12 | 1.21 | 0.97 | 1.51 | 0.093 |
| | gBRCA2pv Exon 11 | 52 | 1.52 | 1.16 | 2.00 | 0.002 | 1.67 | 1.26 | 2.23 | <0.001 |
| | gBRCA2pv outside Exon 11 | 38 | 1.66 | 1.15 | 2.42 | 0.007 | 1.90 | 1.31 | 2.76 | <0.001 |
| g*BRCA*pv domain | Non carriers | 1096 | - | - | - | - | - | - | - | - |
| | g*BRCA1*pv BRCT | 17 | 1.43 | 0.90 | 2.26 | 0.126 | 1.35 | 0.83 | 2.20 | 0.222 |
| | g*BRCA1*pv DBD | 40 | 1.58 | 1.15 | 2.18 | 0.005 | 1.60 | 1.14 | 2.25 | 0.006 |
| | g*BRCA1*pv outside domain | 54 | 1.41 | 1.09 | 1.81 | 0.007 | 1.38 | 1.06 | 1.79 | 0.017 |
| | g*BRCA1*pv RING | 27 | 1.15 | 0.82 | 1.61 | 0.419 | 1.28 | 0.87 | 1.90 | 0.216 |
| | g*BRCA2*pv DBD | 13 | 0.81 | 0.43 | 1.51 | 0.506 | 0.79 | 0.39 | 1.63 | 0.528 |
| | g*BRCA2*pv outside domain | 35 | 2.10 | 1.46 | 3.00 | <0.001 | 2.03 | 1.44 | 2.87 | <0.001 |
| | g*BRCA2*pv RAD51-BD | 39 | 1.37 | 1.01 | 1.85 | 0.04 | 1.58 | 1.14 | 2.21 | 0.006 |
| g*BRCA*pv mutation type | Non carriers | 1096 | - | - | - | - | - | - | - | - |
| | Missense | 27 | 1.20 | 0.85 | 1.71 | 0.297 | 1.20 | 0.80 | 1.79 | 0.372 |
| | Truncating | 192 | 1.48 | 1.27 | 1.73 | <0.001 | 1.58 | 1.35 | 1.86 | <0.001 |
| | SV | 13 | 0.83 | 0.49 | 1.42 | 0.501 | 1.30 | 0.75 | 2.28 | 0.354 |
| | Splice | 6 | 1.24 | 0.64 | 2.43 | 0.521 | 0.87 | 0.45 | 1.67 | 0.670 |

Models were adjusted for FIGO stage, residual disease status, primary tumor site, type of surgery, age at diagnosis (modeled with restricted cubic splines, 3 knots), use of neoadjuvant chemotherapy, tumor grade, and PARP inhibitor use in first-line treatment. AFT models were fitted using a log-logistic distribution. Results are presented as Time Ratios (TR) with 95% confidence intervals (CI). Two-sided *P*-values were derived from Wald tests. A TR > 1 indicates an association with longer time to progression or death, while a TR < 1 reflects shorter survival. The reference group for all comparisons is non-carriers of g*BRCA*pv. Source data are provided as a Source Data file.
*DBD* DNA Binding Domain, *RING* Really Interesting New Gene, RAD51-BD = RAD51 Binding Domain, BRCT = BRCA1 C-Terminal.

protein loss was observed in 13.37% ($n = 88/658$) of patients and was associated with improved survival compared to retained NF1 expression (median OS 4.70 vs. 3.58 years, $P = 0.028$; Supplementary Fig. 8a). Although there were few patients with NF1 protein loss and germline *BRCA1* ($n = 21$) or *BRCA2* ($n = 6$) pathogenic variants, NF1 loss was associated with better survival in g*BRCA2*pv-carriers (median OS 8.05 years NF1 loss vs. 5.72 years NF1 retained) but not in g*BRCA1*pv-carriers (median OS 4.74 years NF1 loss vs. 4.69 years NF1 retained; Supplementary Fig. 8b). NF1 loss also was associated with a longer survival among non-carriers (median OS 5.01 years NF1 loss vs. 3.36 years NF1 retained; Supplementary Fig. 8b).

In the independent OTTA cohort with *NF1* mRNA expression and survival data available ($n = 5666$), low *NF1* expression (lowest quantile) was associated with improved survival compared to high expression (2nd to 5th quantiles) (median OS 4.18 vs. 3.56 years, $P < 0.0001$; Supplementary Figs. 1 and 8c). Consistent with the other cohorts, g*BRCA2*pv-carriers with low *NF1* expression ($n = 36$) showed an improved survival (median OS 6.42 years *NF1* low vs. 5.66 years *NF1* high), while there was no effect in g*BRCA1*pv-carriers (median OS 5.41 years *NF1* low vs. 5.65 years *NF1* high, Supplementary Fig. 8 d).

### *PIK3CA*, *RAD21*, and *MYC* amplifications are associated with short survival in *BRCA2*-deficient HGSC

We found an enrichment of *PIK3CA*, *RAD21*, and *MYC* gene amplifications in *BRCA2*-deficient tumors in patients with short compared to long survival (*PIK3CA*: 5/10, 50% vs 4/24, 16.7%, $P_{adj} = 0.232$; *RAD21*: 5/10, 50% vs 4/24, 16.7%, $P_{adj} = 0.105$; *MYC*: 5/10, 50% vs 3/24, 12.5%,

$P_{adj} = 0.126$, respectively; Fig. 3D, E; Supplementary Notes). Mutual exclusivity analysis showed a co-occurrence of *RAD21* and *MYC* alterations (23/28, $P_{adj} < 0.001$; Supplementary Data 10). Co-amplification of *RAD21* with *MYC* or *PIK3CA* was observed in 20.6% (7/34) and 8.8% (3/34) patients with *BRCA2*-deficiency, respectively (Supplementary Data 10 and 11). *PIK3CA* and *RAD21* mRNA expression was highly correlated with copy number ($P < 0.0001$), and tumors with gene amplification ($\geq 7$ copies) had a significantly higher expression ($P < 0.001$ and $P = 0.02$, respectively) (Supplementary Fig. 9a, b and Supplementary Data 12 and 13). In contrast, *MYC* mRNA expression showed a weaker correlation with copy number ($P = 0.011$), and *MYC* gene amplification ($\geq 7$ copies) was not associated with significantly higher expression ($P = 0.13$; Supplementary Notes, Supplementary Data 14). Patients with combined *BRCA2*-deficiency and *PIK3CA* amplification ($n = 9$, median OS 2.89 years) or *RAD21* amplification ($n = 9$, median OS 2.89 years) had a significantly worse prognosis compared to patients with *BRCA2*-deficient tumors without *PIK3CA* amplification ($n = 25$, median OS 11.92 years) or *RAD21* amplification ($n = 25$, median OS 11.53 years; Supplementary Fig. 9c, d and Supplementary Data 6).

PI-3 kinase pathway activity is thought to contribute to tolerance to genome doubling and *PIK3CA* amplification in whole-genome duplicated tumors is a frequent event in HRD end-stage HGSC[51,67]. The STS *BRCA2*-deficient group was characterized by high ploidy ($P_{adj} = 0.0073$) and whole-genome duplication ($P_{adj} = 0.0404$), in contrast to *BRCA1*-deficient and *BRCA*-proficient tumors where the LTS groups tended to have higher ploidy (Supplementary Fig. 6a). The

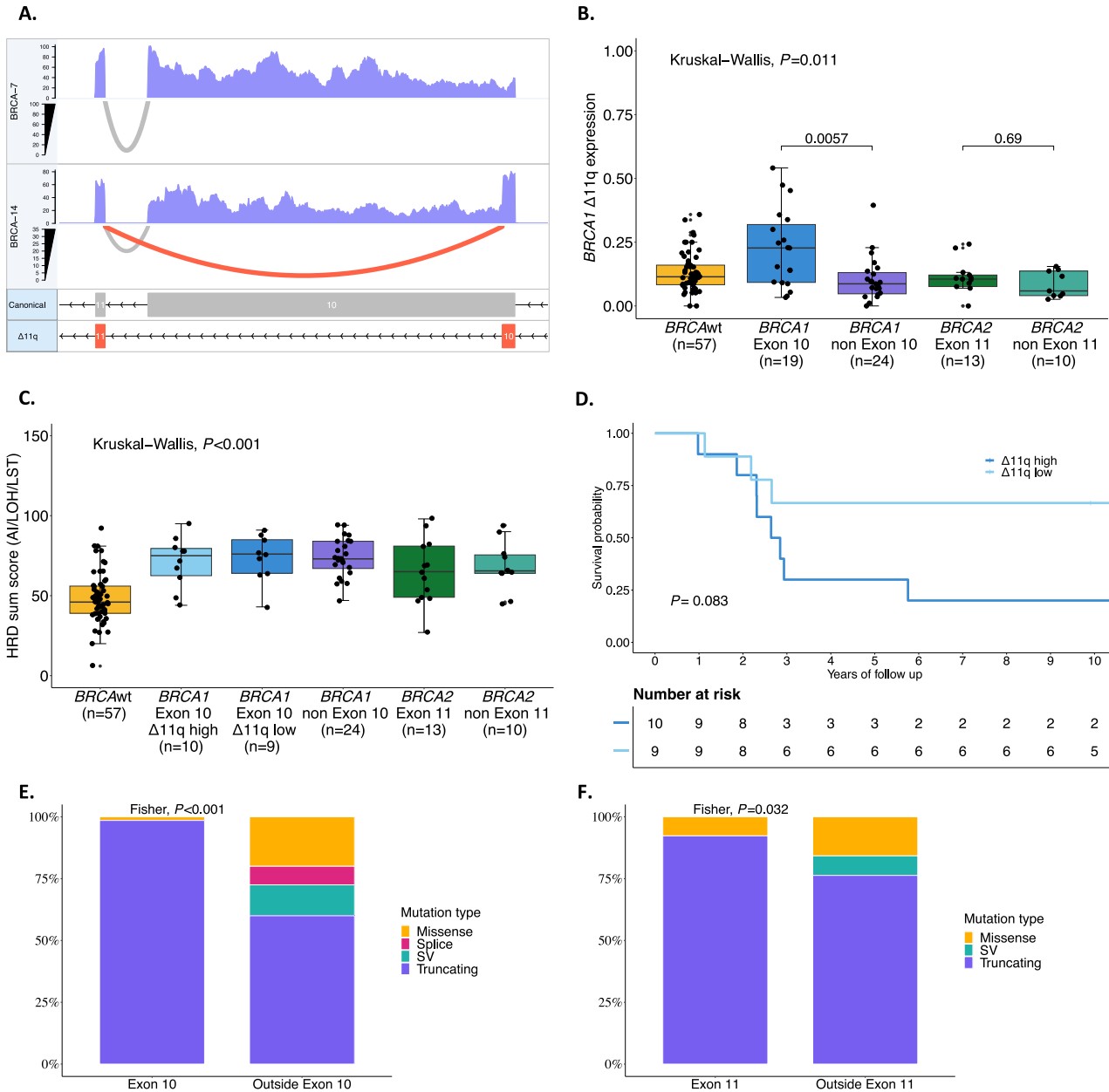

**Fig. 2 | Analysis of pathogenic germline *BRCA1* and *BRCA2* variants and isoform expression on survival in HGSC (MOCOG cohort). A** Illustrates the RNA-seq coverage and splice junction reads across the *BRCA1* gene for two samples (BRCA-7 and BRCA-14). The top and middle panels show the expression levels, with BRCA-7 and BRCA-14 indicating overall expression coverage. The bottom panel depicts the structure of the *BRCA1* isoforms, where the canonical isoform includes exon 10, while the Δ11q isoform excludes it. Gray arcs in the top and middle panels represent splice junction reads supporting the canonical isoform, while red arcs indicate reads supporting the Δ11q isoform. The higher expression of the Δ11q isoform in BRCA-14 compared to BRCA-7 highlights differential splicing events between these samples. **B** Illustrates a comparison of *BRCA1* Δ11q expression among patients (*n* = 123) with mutations in *BRCA1* exon 10 and outside exon 10, *BRCA2* exon 11 and outside exon 11, and patients with *BRCA* wildtype. Box plots display the median (centre line), interquartile range (IQR; 25th–75th percentiles, box bounds), and whiskers extending to the minimum and maximum values within 1.5×IQR. Individual points represent individual patients. *P*-values were calculated using a two-sided Kruskal-Wallis test, with two-sided pairwise Wilcoxon rank-sum tests

performed for post hoc comparisons. **C** Shows HRD sum score distribution among patients (*n* = 123) with mutations in *BRCA1* exon 10 (high and low Δ11q expression) and outside exon 10, *BRCA2* exon 11 and outside exon 11 and *BRCA* wildtype tumors. Box plots display the median (centre line), interquartile range (IQR; 25th–75th percentiles, box bounds), and whiskers extending to the minimum and maximum values within 1.5×IQR. Individual points represent individual patients. *P*-values were calculated using a two-sided Kruskal-Wallis test, with two-sided pairwise Wilcoxon rank-sum tests performed for post hoc comparisons. **D** Kaplan–Meier analysis of overall survival comparing high vs low Δ11q expression (divided by median) in patients (*n* = 19) with a *BRCA1* mutation on Exon 10. *P*-value calculated using a two-sided log-rank test. The distribution of mutation types within *BRCA1* outside exon 10 vs. on exon 10 (*n* = 149 patients) and for *BRCA2* outside exon 11 vs. on exon 11 (*n* = 90 patients) is presented in **E**, **F**, respectively. Group differences were assessed using a two-sided Fisher's exact test. Source data are provided as a Source Data file. BRCAwt = BRCA wildtype, HR Hazard ratio, n Number of patients, SV Structural variants, LST Large scale transitions, LOH Loss of heterozygosity, AI Allelic imbalance.

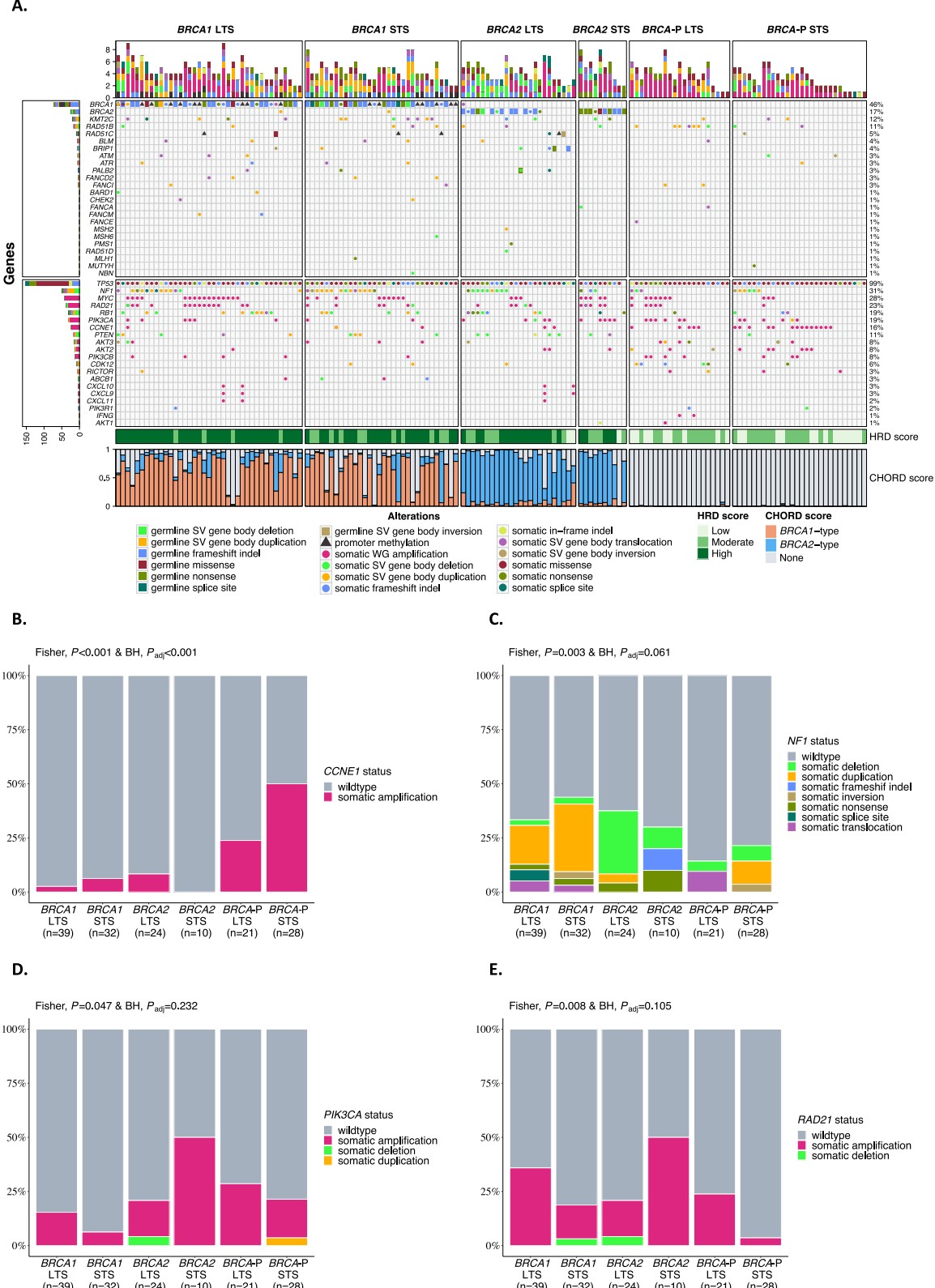

**Fig. 3 | Genetic landscape of HGSC stratified by *BRCA* status and survival (MOCOG cohort). A** Oncoprint showing germline and somatic alterations of homologous recombination (HR) genes and other genes of interest stratified by *BRCA*-status and survival group (*n* = 154 patients). The distribution of the mutation type within the *BRCA* survival group (*n* = 154 patients) is shown for **B** *CCNE1*, **C** *NF1*, **D** *PIK3CA*, and **E** *RAD21*. *P*-values were calculated by a two-sided Fisher's exact test, and *P*-values were adjusted using the Benjamini-Hochberg (BH) procedure (*P*<sub>adj</sub>).

Source data are provided as a Source Data file. *BRCA* status group: Long-term survivor (LTS) = OS > 3 years, Short-term survivor (STS) = OS ≤ 3 years, BRCA-P = BRCA-proficient, HRD score High = ≥ 63 HRD Sum, Moderate = 42–62, Low = ≤41 HRD Sum, HRD Homologous recombination deficiency, CHORD Classifier of HOmologous Recombination Deficiency, SV Structural variant, WG Whole gene, BH Benjamini-Hochberg.

association between *PIK3CA* and survival by *BRCA* status was further corroborated in the OTTA cohort, where gBRCA2pv carriers with high *PIK3CA* RNA expression (highest quantile) had shorter survival relative to their counterparts with low expression (median OS 3.83 vs 7.43 years, *P* < 0.0001; Supplementary Fig. 9e), whereas gBRCA1pv carriers with high *PIK3CA* RNA expression showed improved survival (median OS 7.67 vs 5.23 years). In contrast to *PIK3CA*, there were more modest differences between high and low *MYC* expression groups in gBRCA1pv and gBRCA2pv carriers (median OS 7.18 vs 5.41 years for gBRCA1pv; 5.13 vs 6.41 years for gBRCA2pv, respectively). In *BRCA* wildtype tumors, *MYC* expression demonstrated a stronger correlation with outcome (median OS high 4.14 vs low 3.08 years; Supplementary Notes) compared to *PIK3CA* expression (median OS high 3.21 vs low 3.22 years; Supplementary Fig. 9e).

### Elevated HRD scarring is prognostic for survival in *BRCA*-deficient HGSC

High tumor mutation burden has been shown to be associated with long-term survival in ovarian cancer[22]. However, we found that tumor mutation burden and predicted neoantigen counts were equivalent in *BRCA1*-deficient and *BRCA2*-deficient tumors between STS and LTS groups (Fig. 4A–C, Supplementary Fig. 6a, and Supplementary Data 15). Among various genomic features that were compared between these groups (Supplementary Fig. 6a), the HRD score[27] was elevated in *BRCA1*-deficient tumors with long survival times compared to those with short survival times (*P* = 0.017; Fig. 4D). HRD score is a measure of genomic scarring associated with impaired HR repair, suggesting a more profound inactivation of the HR pathway in patients with good outcome. Retention of the wild-type allele with absence of locus specific LOH has been reported to influence outcomes in gBRCApv-carriers in ovarian and breast cancer[68–71]. However, in our cohort there was only one gBRCA2pv carrier without loss of the wildtype allele (patient BRCA_9; Supplementary Data 7 and Supplementary Notes). Concordantly this tumor was HR-proficient with an HRD score of 27 (HRP ≤ 42 HRD sum score) and CHORD score of 0 (HRP ≤ 0.5 CHORD score), and the patient had short OS (<3 years).

We observed a dynamic range in HRD scores, even among tumors with pathogenic *BRCA* mutations, suggesting a non-equivalence of alterations. The cutoff of the HRD score has been debated, with 42 mainly used in recent clinical trials[72–76], and a more stringent threshold of 63 has been proposed for ovarian cancer[77]. Indeed, patients whose tumors had a high HRD score (≥63) had longer OS (median OS 10 years) compared to those with HRD scores of 42-62 (median OS 2.66 years) and ≤41 (median OS 2.5 years), regardless of *BRCA*-status (*P* = 0.039; Fig. 4E and Supplementary Data 6). Upon applying a threshold of 63 to divide samples into high and low HRD, all *BRCA*-proficient tumors had a low HRD score. Furthermore, patients with *BRCA1*- and *BRCA2*-deficient tumors and HRD scores ≥63 had longer OS compared to patients with lower HRD scores (median OS 6.76 vs. 2.01 years and 11.88 vs. 6.73 years, respectively; Fig. 4F and Supplementary Data 6). Notably, patients with *BRCA1*-deficient tumors with HRD scores <63 had similar OS to patients with *BRCA*-proficient tumors (median OS 2.01 years vs 2.21 years).

Gene Set Enrichment Analysis[78] (GSEA; Methods) revealed distinct patterns of pathway regulation based on HRD scores and *BRCA* status in patients with HGSC. Specifically, pathway activation in *BRCA1*- and *BRCA2*-deficient patients with low HRD (<63) closely resembled those of *BRCA*-proficient patients (Fig. 4G). In contrast, *BRCA1*-deficient patients with high (≥63) HRD scores showed an upregulation of several pathways, including interferon-gamma and inflammatory response. These pathways are primarily involved in host defense and immune surveillance[79], underscoring their potential role in modulating the tumor microenvironment and influencing immune response in patients with *BRCA1*-deficient tumors.

### CD8+PD-1+ T cells are prognostic for survival in gBRCApv-carriers

We considered whether *BRCA*-deficient cases with shorter survival would have fewer mutation-associated neoantigens to drive anti-tumor responses, but there was no difference in neoantigen counts between the STS and LTS groups for both *BRCA1* and *BRCA2* (*P* = 0.51 and *P* = 0.39, respectively; Fig. 4C). Tumor samples from 143 HGSC gBRCApv-carriers were analyzed by multi-color immunofluorescence to determine the epithelial and stromal immune cell densities and their associations with survival groups (Supplementary Fig. 1). Aside from intraepithelial B cells and CD4 + T cells (OR = 1.0), all other immune cell subsets had a positive association with survival (OR < 1.0; Supplementary Table 8). Only intrastromal and intraepithelial CD8 + PD-1 + T cells were significantly more abundant in gBRCApv-carriers with LTS compared to those with STS (*P* = 0.043 and *P* = 0.029, respectively; Supplementary Table 8).

### The mesenchymal features *c-KIT* and mast cells are associated with poor outcome in HGSC

Immune cell abundance was estimated in 154 HGSC tumor samples using CIBERSORTx[80]. Unsupervised clustering of the inferred immune cell densities identified six groups of patients (Fig. 5A, and Supplementary Data 16) associated with differential survival outcomes (*P* = 0.0053; Fig. 5B). The IMMB.1 (*n* = 30) and IMMB.6 (*n* = 25) clusters had exceptionally long survival (median OS 14.87 and 10.45 years, respectively; Supplementary Data 6). The group with the shortest survival (cluster IMMB.5, *n* = 24, median OS 2.03 years) was enriched with activated dendritic cells and resting mast cells, a feature associated with the C1.MES subtype (*P* = 0.0021; Fig. 5C). Although activated dendritic cells were elevated in the IMMB.5 cluster, their association with survival did not remain significant in multivariable Cox regression analysis (Supplementary Fig. 10a). In contrast, resting mast cells emerged as the immune cell type most strongly associated with poor outcome (HR: 1.26, 95% CI 1.06–1.5, *P* = 0.009, Supplementary Fig. 10a). *BRCA1*-deficient tumors in patients with STS had increased expression of the mast cell growth factor receptor *c-KIT* (CD117) compared to those with LTS (*P* = 0.003, $P_{adj}$ = 0.101; Supplementary Fig. 10b). Patients with high *c-KIT* tumor expression had significantly shorter OS than those with low *c-KIT* tumor expression, regardless of *BRCA* and HRD status (HR: 1.71, 95% CI 1.16–2.53, *P* = 0.0071; Supplementary Fig. 10c). The C1.MES subtype showed higher expression of *c-KIT*, together with an upregulation of the epithelial mesenchymal transition (EMT) pathway, compared to the C2.IMM subtype ($P_{adj}$ < 0.001) (Supplementary Fig. 10d, e).

## Discussion

Our study highlights the complexity of survival determinants in patients with HGSC, demonstrating that it is the intersection of multiple factors, including surgical residual disease, immune response, and somatic gene alterations, which may influence outcome rather than *BRCA* mutation status alone. This interplay was particularly apparent in the diminished adverse impact of surgical residual disease in gBRCApv-carriers compared to non-carriers. Previous reports have suggested that surgery in a *BRCA*-deficient setting may have a lesser impact on survival in both first-line and platinum-sensitive setting[33,49,81], indicating that it may be particularly important to achieve complete resection of *BRCA*-proficient tumors. In addition, an exploratory analysis of the PAOLA-1/ENGOT-ov25 trial[82] showed that patients with *BRCA*-proficient tumors classified as higher risk (FIGO stage III with primary cytoreductive surgery and residual disease, or NACT; FIGO stage IV) had notably worse PFS compared to lower-risk patients, while this difference was less pronounced in patients with *BRCA*-deficient tumors. These results emphasize the importance of primary cytoreductive surgery with complete resection for non-carriers, who may also benefit more from secondary cytoreductive

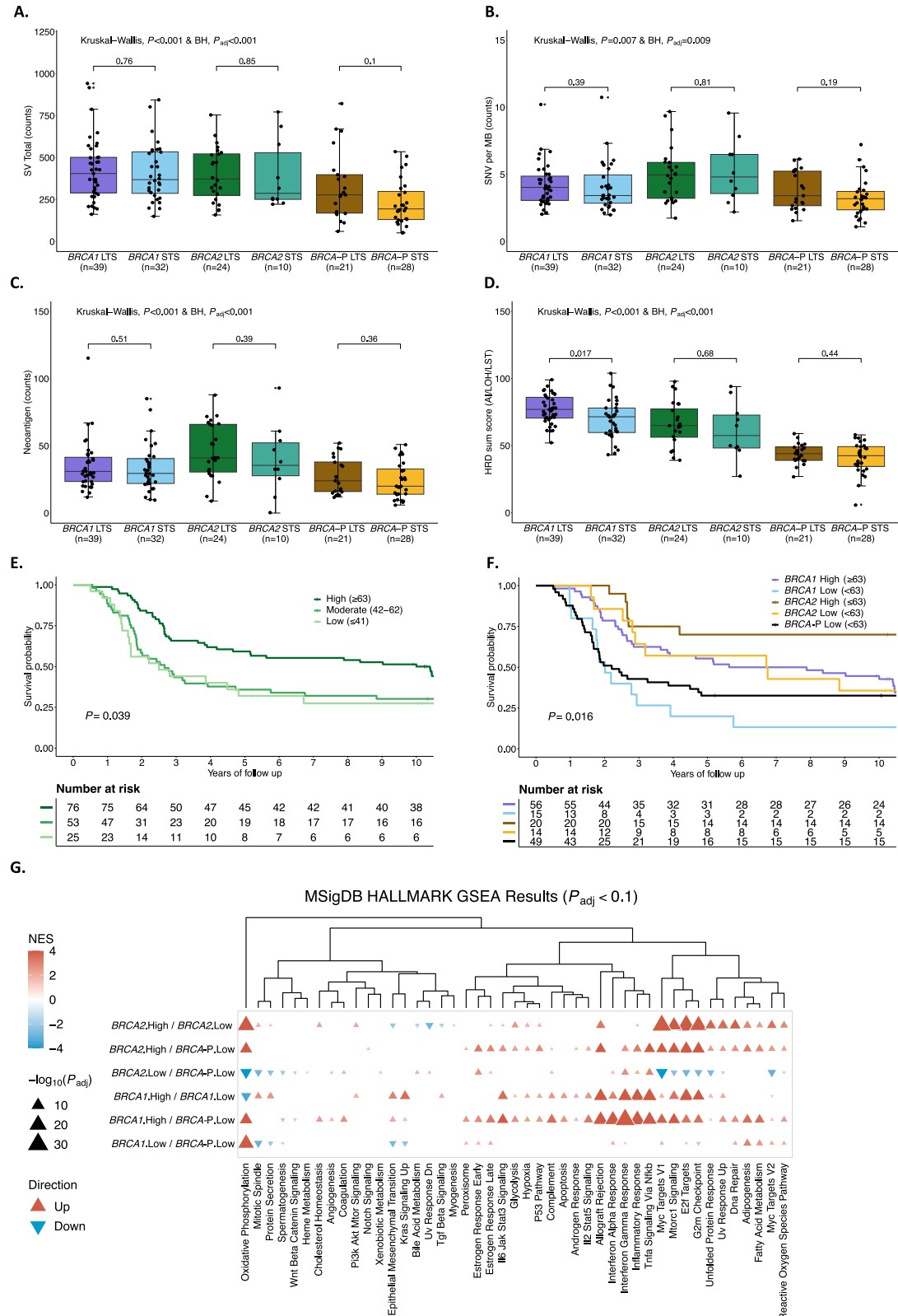

surgery in contrast to gBRCApv-carriers[83]. Equally, it may be that the positive effect of optimal cytoreduction is not as apparent in BRCA carriers, due to the chemotherapy (platinum) sensitivity associated with BRCA-deficiency.

In the current study, the association between NACT and survival appeared to differ by gBRCApv status, with a potential attenuation of survival benefit among gBRCApv-carriers who received NACT.

However, the subgroup analyses by gBRCApv status and treatment type were likely underpowered, limiting definitive conclusions regarding potential interactions. Given the rapid increase in the uptake of NACT in recent years[84], it will be important to determine if patients with BRCA-deficient tumors may be negatively impacted by NACT[85]. Although relapse biopsies were unavailable to assess the emergence of resistance mutations in this study, the acquisition of BRCA reversion

**Fig. 4 | Influence of homologous recombination deficiency in HGSC independent of *BRCA* status (MOCOG cohort).** Comparison of **A** SV total counts, **B** SNV counts per megabase, **C** neoantigen counts, and **D** HRD sum score between *BRCA* survival groups (*BRCA1* = *BRCA1*-deficient; *BRCA2* = *BRCA2*-deficient; *BRCA-P* = *BRCA*-proficient; Long term survivor (LTS) = OS > 3 years; Short term survivor (STS) = OS ≤ 3 years). For **A**–**D** (*n* = 154 patients), box plots represent the median (centre line), with the lower and upper bounds of the box indicating the 25th and 75th percentiles (interquartile range, IQR). Whiskers extend to the minimum and maximum values within 1.5× the IQR. Individual dots correspond to individual tumors. *P* values were calculated using the Kruskal–Wallis test and adjusted for multiple comparisons using the Benjamini–Hochberg (BH) method ($P_{adj}$).
**E** Kaplan–Meier analysis of overall survival stratified by different thresholds of the HRD sum score (High ≥ 63, Moderate 42–62, Low ≤ 42) in 154 patients with *BRCA*-deficient and *BRCA*-proficient HGSC. *P* value calculated by log-rank test.
**F** Kaplan–Meier analysis of overall survival in patients (*n* = 153) with HGSC stratified by *BRCA*-status and high (High ≥ 63) or low (Low < 63) HRD sum score. *P* value

calculated by log-rank test. **G** Clustered heatmap summarizing Gene Set Enrichment Analysis (GSEA) using Hallmark Molecular Signatures Database (MSigDB) gene sets (*n* = 154 patients). Enrichment analysis was performed using FGSEA, and normalized enrichment scores (NES) are shown. Two-sided *P* values were calculated using the FGSEA default permutation-based Monte Carlo method and adjusted for multiple testing using the Benjamini–Hochberg (BH) procedure. Triangle direction and color indicate the direction and magnitude of the NES, while triangle size corresponds to the negative $\log_{10}$ BH-adjusted *P* value ($P_{adj}$). Columns are grouped by BRCA status and HRD score category (*BRCA1*; *BRCA2*; *BRCA*-P, *BRCA*-proficient, High ≥ 63; Low < 63), with enrichment direction defined by the first group listed in each x-axis label. Source data are provided as a Source Data file. SV Structural variants, SNV Single nucleotide variant, MB Megabase, HRD Homologous recombination deficiency, HRP Homologous recombination proficiency, BRCA-P = BRCA-proficient, LST Large scale transitions, LOH Loss of heterozygosity, AI Allelic imbalance.

mutations is frequent[50–52], and it is plausible that reversion events may be more common where chemotherapy commences with a large tumor volume from which resistant clones could emerge under selection[61]. This is especially important in the PARPi era, where the early development of platinum resistance could negatively impact on the potential benefit gained from PARPi treatment. While the impact of NACT on outcomes according to *BRCA* status is not yet known, it is becoming increasingly important to more rapidly determine the *BRCA* and broader HR status of a patient's tumor at diagnosis to make the most informed decisions at primary treatment.

Our study highlights the spectrum of HRD scores seen in patients with *BRCA*-deficient tumors. While all but two exceeded a threshold (>42) required for classification as HRD, the improved OS and PFS seen with a more stringent threshold (≥63) supports that HRD should not be considered a binary classification but rather a continuous variable[26,35]. This finding is consistent with a previous analysis of 537 HGSC cases from The Cancer Genome Atlas which showed that patients with HRD scores ≥63 were associated with better survival outcomes, while those with intermediate (42–62) and low (≤42) HRD scores had overlapping survival curves[77]. It is important to mention that in our study, samples were collected over nearly 20 years, a timeframe that encompasses changes in treatment practices, making it challenging to determine how evolving therapies, particularly the introduction of PARPi, may have influenced outcomes. It is notable that the HRD score threshold of 42 was originally established to predict response to neoadjuvant platinum-containing chemotherapy in patients with breast cancer[86], which tends to have less genomic scarring compared to ovarian cancer[27,77]. As HRD scores ≥63 strongly predicted better outcomes in *BRCA*-deficient HGSC, our findings support the prognostic value of HRD score thresholds. However, it is premature to conclude that a higher threshold should alter therapy selection. To establish this, a comprehensive analysis of maintenance PARPi trials, incorporating HRD scores, would be necessary to confirm their predictive role in guiding treatment decisions. Furthermore, it would be ideal to extend this investigation to include other relevant genomic alterations identified in trial samples to refine patient stratification further. This refinement would help identify patients for whom no maintenance therapy or additional targeted therapy may be more appropriate, while avoiding potentially ineffective treatments for those with lower HRD scores, thereby personalizing therapy to maximize efficacy and minimize unnecessary side effects.

Our analyses corroborated Labidi-Galy et al.'s findings that pathogenic variants in the RAD51-BD of *BRCA2*[46] and the DBD of *BRCA1*[45] are associated with improved outcomes in HGSC. By contrast, alterations outside *BRCA1* exon 10, particularly in the BRCT and RING regions, are not associated with a significantly improved survival compared to non-carriers and in some cases may confer platinum and PARPi resistance[47]. While *BRCA1* exon 10 mutations have been

associated with improved outcomes in multiple studies, including ours, there is evidence that tumors may express the *BRCA1*-Δ11q splice isoform, which bypasses exon 10 mutations and results in a shorter but partially functional protein that is permissive of treatment resistance[45,48]. In our study, patients with a pathogenic *BRCA1* variant in exon 10 and high Δ11q expression had a shorter survival. However, this did not reach statistical significance due to the relatively small sample size for which we had RNA-seq data (*n* = 19 *BRCA1* exon 10 mutated tumors) and we were unable to measure Δ11q expression during or following treatment. This is important because Δ11q expression may increase or fluctuate under the selective pressure of treatment, which would influence treatment response and survival outcomes[87]. Further characterization of whether specific *BRCA* mutations relate to variable expression of *BRCA* splice isoforms in primary tumors is also warranted, as studies have shown that certain splice site mutations are associated with a reduced cancer penetrance phenotype[88], and these could plausibly be associated with a lower HRD score and/or a diminished response to therapy.

CD8 + PD1 + T cells are associated with improved outcomes in ovarian cancer[89], contributing to enhanced anti-tumor immunity. In our analysis, the presence of these cells in tumors was prognostic for survival in g*BRCA*pv-carriers, although to a lesser extent. This suggests that while cytotoxic T-cell activity remains important in *BRCA*-deficient tumors, additional factors may influence survival. Kraya et al.[90] also reported substantial heterogeneity in immune infiltration among *BRCA*-deficient ovarian cancers, identifying immune-high and immune-low subsets characterized by co-occurring genomic alterations such as *PTEN* loss and *BRCA1* promoter hypermethylation. Their findings complement our observations by reinforcing that *BRCA* deficiency alone does not guarantee robust T-cell responses and that tumor-intrinsic immune resistance mechanisms may blunt immunogenicity. Given the established association between *BRCA* and HR status and increased TMB[22], it is possible that immune exhaustion, suppressive signaling or tumor-intrinsic immune resistance pathways may counteract the expected immunogenicity. Intriguingly, *BRCA1*-deficient tumors with high HRD scores had evidence of enhanced immune-related gene transcription. In addition, while our study did not include cigarette smoking in the survival models, smoking has been identified as a potential factor influencing survival in g*BRCA*pv-carriers[91], which may also influence the immune response. Further research into markers of T-cell exhaustion and other immune regulators is needed to better understand the differential immune responses in these patients.

NF1 gene loss-of-function emerged as a good prognostic factor in *BRCA2*-deficient HGSC. Loss-of-function of *NF1* is common in epithelial ovarian cancer with a prevalence of 12–31%[13,20,22,61,92,93]. NF1 inactivation by gene breakage or mutations may contribute to initial good prognosis but later chemoresistance in patients with HGSC and *BRCA*-deficiency[84]. This is consistent with recent findings that deleterious *NF1*

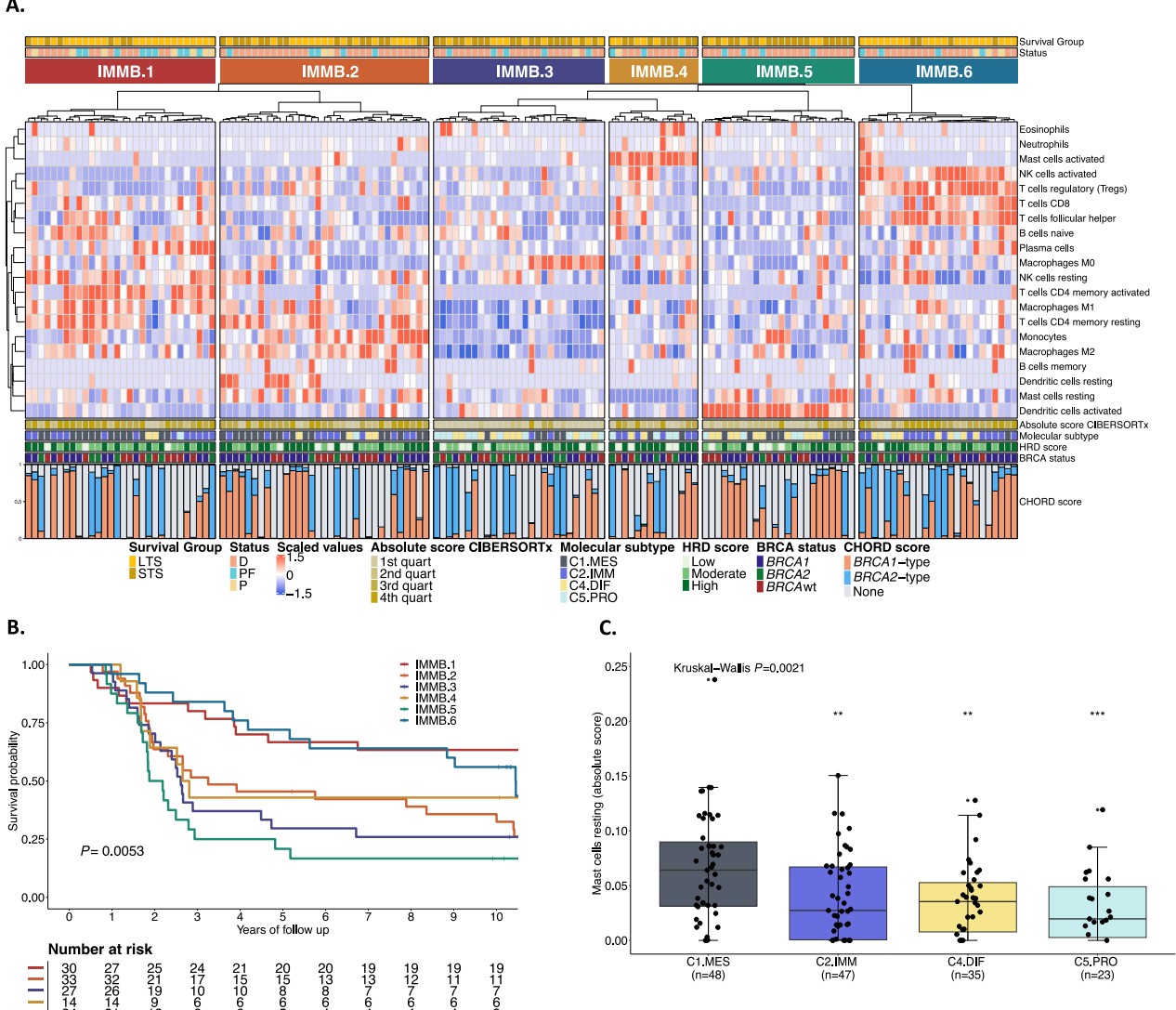

**Fig. 5 | Integration of immune cell profiling by CIBERSORTx and survival analysis in HGSC (MOCOG cohort). A** Summary of the immune cell types arising from the CIBERSORTx analysis from *BRCA*-deficient and *BRCA*-proficient samples (*n* = 153 patients). Tumors fell into 6 major clusters (IMM.1-IMM.6) of immune cell types associated with survival. Each patient is annotated with survival group, status at last follow-up, CIBERSORTx absolute immune scores, molecular subtype, HRD score, *BRCA* status and CHORD score. **B** Kaplan–Meier analysis of overall survival stratified by immune clusters (*n* = 153 patients). *P* value calculated by log-rank test. **C** Box plots summarize the absolute cell enrichment score of mast cells resting markers across the molecular subtype (C1.MES; C2.IMM; C4.DIF; C5.PRO) (*n* = 153 patients); points represent each sample, boxes show the interquartile range (25–75th percentiles), central lines indicate the median, and whiskers show the

smallest/largest values within 1.5 times the interquartile range. Group differences were assessed using a two-sided Kruskal–Wallis test, followed by two-sided pairwise Wilcoxon rank-sum test comparing molecular subtypes (C2.IMM; C4.DIF; C5.PRO) to C1.MES (**\**P* < 0.01, \*\*\**P* < 0.001). Source data are provided as a Source Data file. Survival group: Long term survivor (LTS) = OS > 3 years, Short term survivor (STS) = OS ≤ 3 years, HRD=Homologous recombination deficiency, HRD score: High = ≥ 63 HRD Sum, Moderate = 42-62, Low = ≤41 HRD Sum, Molecular subtypes: C1.MES = C1 mesenchymal subtype, C2.IMM = C2 immunoreactive subtype, C4.DIF = C4 differentiated subtype, C5.PRO = C5 proliferative subtype, Status: D Dead, PF Progression-free, P Progression, IMMB Immune cluster BadBRCA (IMMB.1-IMMB.6).

mutations are associated with improved PFS in ovarian cancer[20] and low mRNA expression of *NF1* predicts longer overall survival[22]. In contrast, *PIK3CA* amplification and high mRNA expression were associated with shorter survival in patients with *BRCA2*-deficient HGSC. As a major regulator of the phosphoinositide 3-kinase (PI3K) pathway, *PIK3CA* activation promotes cell proliferation and survival, especially in genomically unstable cancers[51,67]. Its amplification may enhance tolerance to genome doubling and contribute to the aggressive nature of *BRCA2*-deficient tumors. The contrasting survival outcomes between *PIK3CA* amplification and *NF1* loss-of-function underscore the heterogeneity of HGSC tumors, highlighting the need for personalized therapeutic strategies, even within the *BRCA2*-deficient subgroup.

Our study has limitations that should be considered when interpreting the findings. First, the cohorts span nearly two decades, during which treatment practices evolved substantially; although we adjusted for treatment factors and performed sensitivity analyses excluding PARPi exposure, changes in clinical management may influence outcome estimates. Second, multi-omics profiling was performed on a survival-enriched subset (extreme STS and LTS), which strengthens signal detection but may limit generalizability to the broader HGSC population. Third, several molecular subgroups, such as *BRCA1* exon 10 tumors with high Δ11q expression or *BRCA2*-deficient tumors with *RAD21* amplification, were small, and larger datasets will be required to validate these associations. Finally, while our integrative analyses

identify putative biological mechanisms and candidate modifiers of outcome, functional validation studies are essential to clarify the mechanistic basis and therapeutic relevance of these observations.

In summary, our study demonstrates that survival in *BRCA*-deficient HGSC is determined not by *BRCA* status alone but by the combined influence of HRD severity, co-occurring genomic alterations, and the immune microenvironment. Integrating these features, including an HRD gradient, NF1 loss-of-function, CD8 + PD1 + T-cell activity, and adverse modifiers such as *PIK3CA* and *RAD21* amplification, provides a framework for more refined risk stratification and identifies potential therapeutic targets that may warrant exploration in future clinical trials. These findings support the design of prospective studies incorporating multi-layered genomic and immune profiling to personalize therapy and develop targeted interventions for biologically defined *BRCA*-deficient subgroups.

## Method

### Ethics statement

Written informed consent or an approved waiver of consent was obtained at each participating study site for patient recruitment and the use of samples and linked clinical information (Supplementary Data 17). A waiver of consent applied to cases from five of the 34 OTTA study sites (i.e., the Alberta Ovarian Tumor Types Study, the Brazil Gynecologic Tumor Bank study, the Kliniken Essen-Mitte Department for Gynecology and Gynecologic Oncology, The Netherlands Cancer Institute, and the Vancouver Ovarian Cancer Study) and was granted approval because the studies used anonymized data, had a retrospective design with no active subject recruitment, clinical interventions or impact on patient care, and used archival pathology specimens. Investigations were performed after approval by local human research ethics/institutional review board committees at each site, including the Bioethical Committee of Pomeranian Medical University, the Bioethics and Animal Welfare Committee of the Carlos III Health Institute, the Cambridgeshire 4 Research Ethics Committee, the City of Hope Institutional Review Board (IRB), the Conjoint Health Research Ethics Board, the Duke University Health System IRB, the Landesärztekammer Nordrhein Ethical Review Board, the South East Multi-Centre Research Ethics Committee, the Ethics Committee of the Friedrich-Alexander-University Erlangen-Nuremberg, the Ethics Committee of the Heidelberg University Clinic, the Ethics Committee of the University Hospitals Leuven, the Ethics Committee at the Medical Faculty and at the University Hospital of Tübingen, the Ethics Committee of the Medical Faculty at the University of Heidelberg, the Fred Hutchinson Cancer Research Center IRB, the Health Research Ethics Board of Alberta, the IRB of Cedars-Sinai Medical Center, the IRB Health Research Association and IRB University of Southern California School of Medicine, the Mayo Clinic IRB, the IRB of the Netherlands cancer registry and the IRB of the Netherlands cancer institute, the National Health Service Central Office for Research Ethics Committees and The Joint University College London/University College London Hospital Committee on the Ethics of Human Research, the Peter MacCallum Cancer Centre Human Research Ethics Committee (HREC), the Research Ethics Committee of Hospital das Clínicas of the Ribeirão Preto Medical School, the South Eastern Sydney Local Health District HREC, the St John of God Healthcare HREC and the Women and Newborn HREC, the Swedish Ethical Review Authority, The Duke University Health System IRB for Clinical Investigations, the UCLA IRB, the UK ethics committee, the University of British Columbia - British Columbia Cancer Agency Research Ethics Board, the University of Hawaii Committee on Human Studies, the University of Pittsburgh IRB and Roswell Park Cancer Institute IRB, the US National Cancer Institute special studies IRB, the Western IRB, and the Western Sydney Local Health District HREC. This study was conducted in accordance with the principles of Good Clinical Practice, the Declaration of Helsinki and local regulations.

### Study population

This retrospective, multi-center study included patients diagnosed with HGSC between 2002 and 2019. The Australian Ovarian Cancer Study (AOCS) cohort ($n = 1389$) included all stages (FIGO I-IV), and the Multidisciplinary Ovarian Cancer Outcomes Group (MOCOG) cohort ($n = 154$) was restricted to advanced stage disease (FIGO III and IV; Table 1, Supplementary Fig. 1, and Supplementary Data 17). Patients were categorized based on OS into short (<3 years) and long (≥3 years) OS groups (Supplementary Notes). For multi-omics analysis, 154 patients had fresh-frozen tumor obtained during primary cytoreductive surgery and matched blood samples, or were previously analyzed[22,61]. Findings were validated in an independent HGSC cohort ($n = 5875$) from the Ovarian Tumor Tissue Analysis Consortium (OTTA) for which g*BRCA*pv status was available.

### Molecular data

**Single-nucleotide polymorphism (SNP) arrays.** Tumor and matched normal DNA was analyzed with the Infinium OmniExpress-24 BeadChip arrays as described previously[22]. The concordance of normal and tumor DNA was assessed using HYSYS[94]. Tumor DNA samples with estimated tumor cellularity >40% (determined by qPure[95] and ASCAT[96]) were considered appropriate for whole genome sequencing and methylation arrays.

**Whole genome sequencing (WGS).** For WGS, libraries were generated from tumor and matched normal genomic DNA from peripheral blood mononuclear cells with a minimum base coverage of 60x and 30x, respectively. FASTQ files were assessed for sequencing quality using FASTQC (v0.11.8) and, for contaminants using FastQ Screen[97] (v0.11.4). Adapters, N-content and low-quality bases were trimmed using fastq-mcf (v1.05). Sequenced data was mapped to the human genome reference GRCh37 b37 using the aligner BWA mem[98] (v.0.7.17-r1188). Aligned BAM files per lane were then sorted, merged and duplicates marked using Picard Tools (v.2.17.3). Further processing of the aligned files included base recalibration using GATK Base-Recalibrator (v4.0.10.1). Coverage calculation was performed using GATK DepthOfCoverage (v3.8-1-0-gf15c1c3ef). GATK HaplotypeCaller (v.4.0.10.1) was used on germline BAMs to generate Genomic Variant Call Format (GVCF) files which were used as the Panel of Normals (PoN) in the Mutect2 somatic variant calling workflow. Tumor purity and ploidy were estimated using FACETS[99].

**RNA-sequencing (RNA-seq).** Extracted RNA from tumor tissue samples underwent RNA-seq, with initial quality control checks on raw FASTQ files performed using FastQC[97] (v0.11.8). Adapter, poly (A) tails, N content and low quality base trimming was done using fastq-mcf (v1.05), and contamination was assessed using FastQ Screen[97] (v(0.11.4). Reads were then mapped to the human reference GRCh37.92 using the STAR[100] (v2.6.0b) two-pass method. The mapped reads were then sorted using Picard Tools (v2.17.3). Counts were generated using HTSeq[101] (v0.10.0) on the GRCh37.92 Ensembl release gene annotation. Raw count data was then subsetted to protein coding genes and lowly expressed genes were removed using the following strategy. First, raw counts were converted to CPM (counts per million) and only protein coding genes with a CPM of greater than 0.5 in at least 10 samples were retained. The resulting raw count matrix was then normalized using the trimmed mean of M values (TMM) method using edgeR[102] (v3.28.1). Batch effects were removed using limma's[103] (v3.48.2) removeBatchEffect function. Batch effect removal was done by applying batch correction on the library type (stranded/unstranded) while preserving the survival group (long/short).

**Methylation arrays.** The generation and processing of methylation array data was performed as previously described by Garsed et al.[22]. Briefly, initial quality control was performed by QuantiFluor

(Promega). Subsequently, 500 ng tumor DNA was converted using the EZ DNA Methylation kit (Zymo Research) and analyzed using the Infinium MethylationEPIC BeadChip arrays. The R package minfi[104] (v1.32.0) was then used for quality control assessment and processing of the methylation data as previously described[22].

**Immunofluorescence (IF) data.** Tissue microarrays (TMAs) were constructed from formalin-fixed paraffin-embedded (FFPE) blocks of tumor tissue and stained by IF with two panels of antibodies against immune markers of interest. Panel 1 detected CD3, CD8, CD20, FOXP3 and CD79; panel 2 detected CD3, CD8, PD-1, PD-L1 and CD68. Both panels also detected pan-cytokeratin to identify tumor epithelium. Detailed antibody information is provided in the Supplementary Data 18. Automated cell scoring, including separation of epithelial and stromal regions, was performed using QuPath (v0.2m2), with extensive manual training and validation. CD4 + T cells were defined as CD3 + CD8- cells, as previously[105].

**Immunohistochemistry (IHC).** Sections of 4 μm thickness were cut from previously constructed TMAs of FFPE tumor samples. Deparaffinized sections were stained with the C-terminal NF1 antibody (clone NFC, SIGMA #MABE1820; St. Louis, MO, USA) using our previously described protocol on a DAKO Omnis platform: 30 min of pretreatment heat-induced antigen retrieval in Tris-EDTA buffer, pH = 9.0; primary antibody incubation for 1 h at dilution 1/50, 10 min of a mouse linker, and 30 min for the peroxidase labeled Dako EnVision +polymer-based detection system (Dako protocol 1 h-10M-30, Agilent, Santa Clara, CA, USA)[93]. Samples were scored as follows: inactivated (loss of expression with retained internal control), normal retained expression, subclonal loss, uninterpretable (loss of tumor expression but no internal control present), and exclude (no tumor in core) (Supplementary Notes).

**mRNA expression data by NanoString.** Tumor mRNA expression data for genes of interest (*NF1, PIK3CA, c-KIT*, and *RB1*) and transcriptional molecular subtypes in the OTTA cohort were determined using NanoString, as previously described[106,107].

## Measurements

**Variant detection and annotation.** Variant calling was performed for:
1) germline base substitution and INDEL variants by VarDictJava[108] (v1.5.7 with −r = 2 −Q = 10 −f = 0.1).
2) somatic base substitution and INDEL variants using four separate variant callers as follows: by Mutect2[109] (v4.0.11.0 with defaults), VarDictJava[108] (v1.5.7 with −r = 2 −Q = 10 −V = 0.05 −f = 0.01), Strelka2[110] (v2.9.9 with defaults), and VarScan2[111] (SAMtools[112]) v1.9 for mpileup and VarScan2 v2.4.3 with -min-coverage 7 -min-var-freq = 0.05 -min-freq-for-hom = 0.75 -*p* -value = 0.99 -somatic-p-value = 0.05 -strand-filter =0). Variant calls were decomposed and normalized using vt[113] GATKs ReadBackPhasing tool (v3.8-1-0-gf15c1c3ef with -phaseQualityThresh = 10 – enableMergePhasedSegregatingPolymorphismsToMNP -min_base_quality_score =10 -min_mapping_quality_score = 10 -maxGenomicDistanceForMNP = 2) was applied on the passing variants per tool to combine contiguous SNVs to MNVs (multi-nucleotide variants). GATK's CombineVariants (v3.8-1-0-gf15c1c3ef with -genotypeMergeOptions UNIQUIFY -priority Strelka2, Mutect2, VarScan2, VarDictJava) was used to merge the variant calls from all four callers into a consensus variant call set. The resulting variant call format (VCF) file was once again decomposed and normalized using vt. Forward and reverse strand counts for the reference and alternate alleles were calculated using bam-readcount (v0.8.0). Finally, all variants were annotated for Duke and DAC blacklisted regions. Any variants that were passed in at least two callers, had at least one variant read in

each strand, and were not in the database of FrequentLy mutAted GeneS (FLAGS)[114] or the Duke and DAC blacklist regions were deemed high-confidence.
3) structural variants (SV) using four separate callers Manta[115] + BreakPointInspector (v1.5.0), GRIDSS[116] (v2.0.1), Smoove (v0.2.2) and SvABA[117] (v134). The SV calls were split into germline and somatic VCFs per caller. The findBreakpointsOverlaps method of the R library StructuralVariantAnnotation (v1.3.1) with a value of 10 for the 'maxgap' parameter was used to intersect common breakpoints between the callers. SVs were annotated to constituent types (duplication, deletion, inversion or translocation) using a simple annotation script provided by the GRIDSS tool. High-confidence SVs were categorized as those called by two or more callers.
4) copy number variations (CNV) detection by FACETS[99] and cnv_facets (v0.13.0) as described previously[22].

The detected variants were filtered for variants with a high probability of pathogenicity as described in detail before[22].

**Mutation burden and downsampling.** We downsampled the higher coverage tumor BAM files using Picard DownsampleSam (v2.17.3) to achieve balanced median coverage sequencing batches, to compare mutation burden across samples with inconsistent coverage[22]. The median coverage of the International Cancer Genome Consortium (ICGC) tumors was 52.15x, the MOCOG tumors was 77.81x and the short survival *BRCA* dataset tumors was 64.98x. So, to get the same median coverage across the three batches we downsampled the MOCOG and short survival *BRCA* dataset tumors to the ICGC median by specifying downsampling fractions of 0.67 and 0.8 respectively. See Supplementary Table 19 for details on the tumor sample coverage before and after downsampling and the number of SNVs, MNVs, indels and SVs called after downsampling.

**Neoantigen prediction.** Neoantigen prediction was performed as previously reported by Garsed et al.[22]. Briefly, HLA-VBSeq[118] (v11_22_2018) was used to generate HLA types which were then used to identify and construct neoantigen using pVACtools[119] pVACseq (v1.3.5).

**Homologous recombination deficiency (HRD).** HRD status was determined using (1) scarHRD[120], which uses loss of heterozygosity (LOH), telomeric allelic imbalance (TAI), and large scale state transition (LST) in tumor genomes to generate a HRD sum score, and (2) CHORD (Classifier of Homologous Recombination Deficiency)[63], which uses specific base substitution, indel and structural rearrangement signatures detected in tumor genomes to generate *BRCA1*-type and *BRCA2*-type HRD scores.

**RNA-seq data analysis.** Raw count data was subsetted to protein coding genes and lowly expressed genes were removed using the following strategy. First, raw counts were converted to CPM (counts per million) and only protein coding genes with a CPM of greater than 0.5 in at least 10 samples were retained. The resulting raw count matrix was then normalized using the trimmed mean of M values (TMM) method using edgeR[102] (v3.28.1). Batch effects were removed using limma's[103] (v3.48.2) removeBatchEffect function. Batch effect removal was done by applying batch correction on the library type (stranded/unstranded) while preserving the survival group (long-term/short-term).

**RNA differential expression and pathway analysis by grouping.** *Groupings:* For differential expression and pathway analysis, various groupings were used alone or in combination, namely (1) *BRCA*-deficiency status, (2) HRD groups, survival groups, and (3) molecular subtypes (Supplementary Notes).

*Differential expression analysis:* To identify differentially expressed protein-coding genes between the comparison groups of interest, DESeq2 (v1.26.0)[121] was applied. Raw counts were filtered to remove low expressed genes prior to analysis and batch effects were accounted for in the model[22].

*Gene Set Enrichment Analysis (GSEA):* FGSEA v1.15.1 was used to calculate gene set enrichment across the comparison groups. *P*-values obtained from DESeq2 were transformed to signed *P*-values and then sorted and fed into FGSEA to generate enrichment scores and FDR-adjusted *P*-values across the Hallmark gene sets in the MSigDB database49 (v7.4) via its function fgseaMultilevel (minSize=15, maxSize = 500, gseaParam = 0, eps = 0)[22].

**CIBERSORTx.** CIBERSORTx analysis was performed as previously described[22]. Briefly, CIBERSORTx[80] with the LM22 matrix was used on RNA-seq data for immune cell deconvolution. Immune clusters were then defined using the pam clustering algorithm and pearson distance metric on the absolute cell abundances using ConsensusClusterPlus[122] (Supplementary Notes).

**Immunofluorescence.** Data were categorized based on epithelial content, measured directly by pan-cytokeratin positivity and cell morphology (assessed by automated image analysis). Epithelium-negative, cellular (i.e., non-necrotic) tumor regions were defined as stroma. Immunomarker density (D; cells/mm$^2$) for a given marker was calculated separately for epithelial and stromal compartments. For cases with multiple cores, the epithelial area was taken as the sum of all their individual TMA epithelial areas and similarly for the stromal area. We categorized marker D values into quartiles (separately for epithelial and stromal markers) to provide categorical comparisons for ease of interpretation of the odds ratios (ORs). Conditional logistic regression models were fitted for the long survival group vs short survival group. Logistic regression analyses were performed with the quartile values (scored as 1, 2, 3, 4). Immune clusters were then defined using the pam algorithm and pearson distance metric on the immune cell type densities using ConsensusClusterPlus[122].

**Statistical analyses.** Continuous variables were compared between groups using the Kruskal-Wallis test and the difference between proportions of categorical data were assessed using the Chi-squared or Fisher's exact test. Correlations between continuous variables were assessed using Spearman correlation. Benjamini-Hochberg adjusted *P*-values are reported as *P*$_{adj}$ to account for multiple testing. Median PFS and OS were estimated using the Kaplan–Meier method and survival distribution were compared using the log-rank (Mantel-Cox) test.

For the AOCS cohort, univariable and multivariable survival analyses were performed using Accelerated Failure Time (AFT) models[56] with a log-logistic distribution to evaluate associations between clinical and molecular variables and time-to-event outcomes. Results were reported as Time Ratios (TR) with 95% confidence intervals (CI), where a TR > 1 indicates longer time to progression or death, and a TR < 1 indicates shorter survival. Wald tests were used to compute *P*-values for individual covariates and interaction terms. Age at diagnosis was modeled using restricted cubic splines with three knots to allow for potential non-linear effects. Model assumptions were assessed using quantile-quantile plots of deviance residuals and Cox-Snell residuals to evaluate overall model fit. The Akaike Information Criterion (AIC) was used to compare alternative parametric distributions and confirm the suitability of the log-logistic model[123].

For survival analyses of the OTTA cohort, Cox proportional hazards models were applied. Left truncation was used to account for delayed study enrollment at some sites, and follow-up time was right-censored at 10 years from diagnosis to minimize the influence of non-ovarian cancer-related deaths. *P*-values from Cox models correspond to Wald and log-rank tests. The proportional hazards assumption was assessed using the Grambsch-Therneau test based on scaled Schoenfeld residuals and further evaluated through graphical inspection of Schoenfeld residual plots[123,124].

All statistical tests were two sided and considered significant when *P* < 0.05 or *P*$_{adj}$ < 0.1. All analyses were performed using the statistical software R version 4.1.3[125].

**Reporting summary**
Further information on research design is available in the Nature Portfolio Reporting Summary linked to this article.

# Data availability

Short survival B*RCA dataset:* WGS, RNA-seq and SNP array data from short-term survivors generated as part of the current study have been deposited in the European Genome-phenome Archive (EGA) repository (https://ega-archive.org) under accession code EGAS00001008059. WGS and RNA-seq data are available as raw FASTQ files for each sample type (tumor/normal) and SNP array data are available as raw signal intensity files in text format for each sample type (tumor/normal). Controlled access to patient sequence data can be gained for academic use via the EGA, typically for a period of five years from the date the data transfer agreement is fully executed. Information on how to apply for access is available at the EGA under accession code EGAS00001008059. Responses to data requests will be provided within ten business days. The raw methylation data sets have been submitted to the Gene Expression Omnibus (GEO; https://www.ncbi.nlm.nih.gov/geo/) under accession code GSE292140 with no access restrictions. *ICGC dataset:* Previously published WGS and RNA-seq data generated as part of the ICGC Ovarian Cancer project[61] are available from the EGA repository as a single bam file for each sample type (tumor/normal), under the accession code EGAD00001000877. Due to the sensitive nature of these patient datasets, access is subject to approval from the ICGC Data Access Compliance Office, an independent body who authorizes controlled access to ICGC sequencing data. ICGC SNP array and methylation data sets have been deposited into GEO under accession code GSE65821, without access restrictions. ICGC gene count level transcriptomic data has been deposited into the GEO under accession code GSE209964. *MOCOG dataset:* WGS, RNA-seq and SNP array data from long-term survivors generated as part of the MOCOG study[22] have been deposited in the EGA repository under accession code EGAS00001005984. WGS and RNA-seq data are available as raw FASTQ files for each sample type (tumor/normal) and SNP array data are available as raw signal intensity files in text format for each sample type (tumor/normal). Controlled access to patient sequence data can be gained for academic use via the EGA, typically for a period of five years from the date the data transfer agreement is fully executed. Information on how to apply for access is available at the EGA under accession code EGAS00001005984. Responses to data requests will be provided within ten business days. The MOCOG cohort raw methylation data sets have been submitted to the GEO under accession code GSE211687, with no access restrictions. Uniformly processed somatic variant data from the ICGC, MOCOG, and short survival *BRCA* cohorts is deposited in Synapse under accession code syn65463502 and processed methylation and expression data from all cohorts has been submitted into the GEO under accession codes GSE292140 and GSE292142, without access restrictions. *OTTA dataset:* The data underlying the figures and tables are provided in the Source Data file. Population frequencies of genetic variants can be accessed via the Genome Aggregation Database (gnomAD) at https://gnomad.broadinstitute.org/. Supporting evidence for pathogenicity of genomic alterations can be accessed via ClinVar (https://www.ncbi.nlm.nih.gov/clinvar/), BRCA Exchange (https://brcaexchange.org/) and the *TP53* Database (https://tp53.cancer.gov/). The Ensembl ranked order of severity of variant consequences is available at: https://www.ensembl.org/info/genome/variation/prediction/predicted_data.html.

Mutational signature reference databases can be accessed via COSMIC (https://cancer.sanger.ac.uk/signatures/) and Signal (https://signal.mutationalsignatures.com/). The LM22 signature matrix used for immune cell deconvolution can be downloaded here: https://cibersortx.stanford.edu/. MSigDB hallmark gene sets can be accessed here: https://www.gsea-msigdb.org/gsea/msigdb/. Illumina methylation probes that were filtered out due to poor performance (e.g., cross reactive or non-specific probes) can be found here: https://github.com/sirselim/illumina450k_filtering. Germline polymorphic sites for reference and variant allele read counts used in FACETS analysis can be found at https://ftp.ncbi.nih.gov/snp/organisms/human_9606_b151_GRCh37p13/VCF/common_all_20180423.vcf.gz. The GTF used for annotation and RNA-seq counts is available here: https://ftp.ensembl.org/pub/grch37/release-92/. All other data are available within the article and its Supplementary and Source Data files. Source data are provided with this paper.

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

## Acknowledgements

We thank A. Freimund, R. Lupat, J. Ellul, and the Peter MacCallum Cancer Centre Research Computing Facility for their contributions to the study. This work was supported by the National Health and Medical Research Council (NHMRC) of Australia (GNT1186505 and GNT2029088), the US Army Medical Research and Materiel Command Ovarian Cancer Research Program (Award No. W81XWH-16-2-0010 and W81XWH-21-1-0401), the National Institutes of Health (NIH) (R21-CA267050, K07-CA080668, R01-CA95023, R01-CA248288, P50-CA136393, P30-CA015083, MO1-RR000056), the Swiss National Foundation (P500PM_20726); Bangerter-Rhyner Stiftung (0297); Margarete and Walter Lichtenstein-Stiftung; and Freie Gesellschaft Basel. The Gynecological Oncology Biobank at Westmead was funded by the NHMRC (ID310670, ID628903); the Cancer Institute NSW (12/RIG/1-17, 15/RIG/1-16); the Department of Gynaecological Oncology, Westmead Hospital; and acknowledges financial support from the Sydney West Translational Cancer Research Centre, funded by the Cancer Institute NSW (15/TRC/1-01). Direct funding for the generation of the NanoString data for OTTA was provided by the NIH (R01-CA172404, and R01-CA168758), the Canadian Institutes for Health Research (Proof-of-Principle I program) and the United States Department of Defense Ovarian Cancer Research Program (OC110433). T.A.Z. is supported by the Swiss National Foundation Return CH Postdoc.Mobility (P5R5PM_222151). D.W.G. is supported by a Victorian Cancer Agency/Ovarian Cancer Australia Low-Survival Cancer Philanthropic Mid-Career Research Fellowship (MCRF22018) and the Ovarian Cancer Research Foundation (2025/OCRF0071). S.J.R. is supported by the NHMRC (2009840). M.J.G is supported by the Ministerio de Ciencia, Innovación y Universidades (MICIU)/AEI/10.13039/501100011033 and ERDF, EU (Project PID2023-151298OB-I00). A.O. is partially funded by Ministerio de Ciencia e Innovación, Instituto de Salud Carlos III (PI23/01235) supported by FEDER funds and the Spanish Network on Rare Diseases (CIBERER). K.M.D., T.P.C., and G.L.M. were supported by awards from the Uniformed Services University of the Health Sciences and the Defense Health Program to the Henry M Jackson Foundation (HJF) for the Advancement of Military Medicine Inc. to the Gynecologic Cancer Center of Excellence Program including HU0001-16-2-0006 (PIs: Chad A. Hamilton and G. Larry Maxwell), HU0001-19-2-0031, HU0001-20-2-0033, and HU0001-21-2-0027 (PIs: Yovanni Casablanca and G. Larry Maxwell), HU0001-22-2-0016 and HU0001-23-2-0038 (PIs: Neil T. Phippen and G. Larry Maxwell), as well as HU0001-23-2-0038 and HU0001-24-2-0047 (PIs Christopher M Tarney and G. Larry Maxwell). T.V.G. is a Senior Clinical Investigator of the Fund for Scientific Research-Flanders (FWO Vlaanderen 18B2921N). A.DeF. is supported by the NHMRC (2033042). The AOV study was funded by the Canadian Institutes for Health Research (MOP-86727). The Generations Study was funded by Breast Cancer Now and the United Kingdom National Health Service funding to the Royal Marsden/Institute of Cancer Research. The UK Ovarian Cancer Population study (UKOPS) was funded by The Eve Appeal (The Oak Foundation) with contribution to authors' salary through MRC core funding MC_UU_00004/01 and the NIH Research University College London Hospitals Biomedical Research Centre. The contents of the published material are solely the responsibility of the authors and do not reflect the views of the NHMRC, NIH, and other funders.

## Author contributions

T. A. Z.: Conceptualization, data curation, formal analysis, funding acquisition, validation, investigation, visualization, methodology, writing–original draft, writing–review and editing. S.F.: Conceptualization, data curation, validation, methodology, writing–original draft, writing–review and editing. A.P.: Conceptualization, data curation, formal analysis, validation, investigation, visualization, methodology, writing–original draft, writing–review and editing. D.A.: Data curation, validation, investigation, visualization, methodology, writing–original draft, writing–review and editing. M.W.J.: Formal analysis, validation, investigation, visualization, methodology, writing–original draft, writing–review and editing. L.T.: Formal analysis, validation, investigation, visualization, methodology, writing–original draft, writing–review and editing. A.F.: Conceptualization, data curation, investigation, writing–review and editing. C.M.L.: Formal analysis, validation, investigation, methodology, writing–review and editing. C.J.K.: Resources, data curation, methodology, writing–original draft, writing–review and editing. A.B.: Resources, writing–review and editing. N.S.M.: Resources, writing–review and editing. K.M.: Resources, writing–review and editing. P.H.: Data curation, formal analysis, validation, investigation, methodology, writing–review and editing. J.A.: Resources, writing–review and editing. A.C.A.: Resources, writing–review and editing. G.A-Y.: Resources, writing–review and editing. M.W.B.: Resources, writing–review and editing. A.B.: Resources, writing–review and editing. C.B.: Resources, writing–review and editing. F.B.: Resources, writing–review and editing. C.B.: Resources, writing–review and editing. J.B.: Resources, writing–review and editing. A.H.B.: Resources, writing–review and editing. M.E.C.: Resources, writing–review and editing. A.C-J.: Resources, writing–review and editing. D.S.C.: Resources, writing–review and editing. E.L.C.: Resources, writing–review and editing. A.C-G.: Resources, writing–review and editing. P.C.: Resources, writing–review and editing. K.L.C-H.: Resources, writing–review and editing. C.C.: Resources, writing–review and editing. K.M.D.: Resources, writing–review and editing. C.D.: Resources, writing–review and editing. T.D.: Resources, writing–review and editing. A.B.E.: Resources, writing–review and editing. E.E.: Resources, writing–review and editing. J.E.: Resources, writing–review and editing. T.E.: Resources, writing–review and editing. R.F.: Resources, writing–review and editing. A.F.: Resources, writing–review and editing. M.G-C.: Resources, writing–review and editing. A.G-M.: Resources, writing–review and editing. P.G.: Resources, writing–review and editing. R.G.: Resources, writing–review and editing. P.H.: Resources, writing–review and editing. A.D.H.: Resources, writing–review and editing. A.H.: Resources, writing–review and editing. S.H.: Resources, writing–review and editing. B.Y.H.: Resources, writing–review and editing. A.H.: Resources, writing–review and editing.

S.H.: Resources, writing–review and editing. D.G.H.: Resources, writing–review and editing. M.J-L.: Resources, writing–review and editing. M.E.J.: Resources, writing–review and editing. E.K.: Resources, writing–review and editing. E.K.: Resources, writing–review and editing. T.K.: Resources, writing–review and editing. F.K.F.K.: Resources, writing–review and editing. G.K.: Resources, writing–review and editing. R.F.P.M.K.: Resources, writing–review and editing. J.K.: Resources, writing–review and editing. D.L.: Resources, writing–review and editing. C-H.L.: Resources, writing–review and editing. J.L.: Resources, writing–review and editing. S.C.Y.L.: Resources, writing–review and editing. Y.L.: Resources, writing–review and editing. A.L.: Resources, writing–review and editing. J.L.: Resources, writing–review and editing. L.L.: Resources, writing–review and editing. J.L.: Resources, writing–review and editing. C.M.: Resources, writing–review and editing. I.A.M.: Resources, writing–review and editing. M.M.: Resources, writing–review and editing. G.S.N.: Resources, writing–review and editing. N.N.: Resources, writing–review and editing. A.O.: Resources, writing–review and editing. S.O.: Resources, writing–review and editing. A.O.: Resources, writing–review and editing. C.M.Q.: Resources, writing–review and editing. G.RM.: Resources, writing–review and editing. I.R-C.: Resources, writing–review and editing. C.R-A.: Resources, writing–review and editing. P.R.: Resources, writing–review and editing. M.R.: Resources, writing–review and editing. S.G.S.: Resources, writing–review and editing. S.S.: Resources, writing–review and editing. M.J.S.: Resources, writing–review and editing. H-P.S.: Resources, writing–review and editing. G.S.S.: Resources, writing–review and editing. L.S.: Resources, writing–review and editing. C.J.R.S.: Resources, writing–review and editing. A.T.: Resources, writing–review and editing. A.T.: Resources, writing–review and editing. C.M.T.: Resources, writing–review and editing. S.E.T.: Resources, writing–review and editing. K.K.V.: Resources, writing–review and editing. M.A.A.: Resources, writing–review and editing. T.G.: Resources, writing–review and editing. E.N.: Resources, writing–review and editing. L.W.: Resources, writing–review and editing. A.E.W-H.: Resources, writing–review and editing. C.W.: Resources, writing–review and editing. C.W.: Resources, writing–review and editing. J.W.: Resources, writing–review and editing. N.W.: Resources, writing–review and editing. L.R.W.: Resources, writing–review and editing. S.J.W.: Resources, writing–review and editing. B.W.: Resources, writing–review and editing. M.S.A.: Resources, writing–review and editing. A.B.: Resources, writing–review and editing. F.J.C-R.: Resources, writing–review and editing. P.A.C.: Resources, writing–review and editing. T.P.C.: Resources, writing–review and editing. P.C.: Resources, writing–review and editing. J.A.D.: Resources, writing–review and editing. P.A.F.: Resources, writing–review and editing. R.T.F.: Resources, writing–review and editing. M.J.G.: Resources, writing–review and editing. S.A.G.: Resources, writing–review and editing. M.T.G.: Resources, writing–review and editing. J.G.: Resources, writing–review and editing. H.R.H.: Resources, writing–review and editing. F.H.: Resources, writing–review and editing. H.M.H.: Resources, writing–review and editing. B.Y.K.: Resources, writing–review and editing. L.E.K.: Resources, writing–review and editing. G.L.M.: Resources, writing–review and editing. U.M.: Resources, writing–review and editing. F.M.: Resources, writing–review and editing. S.L.N.: Resources, writing–review and editing. J.M.S.: Resources, writing–review and editing. A.S.: Resources, writing–review and editing. A.J.S.: Resources, writing–review and editing. I.V.: Resources, writing–review and editing. A.H.W.: Resources, writing–review and editing. J.D.B.: Resources, writing–review and editing. P.D.P.P.: Resources, writing–review and editing. C.L.P.: Resources, writing–review and editing. M.C.P.: Resources, writing–review and editing. E.L.G.: Resources, writing–review and editing. S.J.R.: Conceptualization, resources, data curation, supervision, funding acquisition, validation, writing–original draft, project administration, writing–review and editing. M.K.: Conceptualization, resources, data curation, investigation, formal analysis, validation, visualization, supervision, methodology, writing–original draft, writing–review and editing. B.N.: Resources, data curation, investigation, formal analysis, validation, visualization, methodology, writing–review and editing. A.DF.: Conceptualization, Resources, writing–review and editing. M.L.F.: Conceptualization, Resources, writing–review and editing. D.D.L.B.: Conceptualization, resources, supervision, funding acquisition, writing–original draft, writing–review and editing. D.W.G.: Conceptualization, resources, data curation, formal analysis, supervision, funding acquisition, validation, investigation, visualization, methodology, writing–original draft, project administration, writing–review and editing.

## Competing interests

T.A.Z. reports personal consulting fees from AbbVie that are outside the submitted work. D.D.L.B. reports research support grants from AstraZeneca, Roche-Genentech and BeiGene paid to institution outside the submitted work; also, personal consulting fees from Exo Therapeutics that are outside the submitted work. G.A.-Y. reports research support grants from AstraZeneca and Roche-Genentech paid to institution outside the submitted work; also, personal consulting fees from Incyclix Bio that are outside the submitted work. A.DeF. reports research support from AstraZeneca and Illumina. N.N. reports research support from Illumina. P.A.C. reports speakers' honoraria from AstraZeneca, Merck Sharpe and Dohme, and GlaxoSmithKline, and personal consulting fees from Astra Zeneca outside the remit of the submitted work. U.M. and A.G.M. report personal consulting fees from Mercy BioAnalytics Ltd and research support grants from Intelligent Lab on Fiber, RNA Guardian, and MercyBio Analytics that are all outside the remit of the submitted work. E.L.C. reports research support from AstraZeneca paid to institution outside the submitted work and speakers' honoraria from AstraZeneca and GSK. S.E.T reports consulting fees from AstraZeneca and IntegraConnect outside the submitted work. P.H. reports honoraria and consulting fees from Amgen, Astra Zeneca, GSK, Roche, Immunogen, Sotio, Stryker, ZaiLab, MSD, Clovis, Miltenyi, Eisai, Mersana, Exscientia, Daiichi Sankyo, Karyopharm, Abbvie, Novartis, Corcept, BionTech, Zymeworks and Research funding (Institutional) from Astra Zeneca, Roche, GSK, Genmab, Immunogen, Seagen, Clovis, Novartis, Immatics, Abbvie, MSD. I.V. has participated in consulting advisory boards for Akesobio, Bristol Myers Squibb, Eisai, F. Hoffmann-La Roche, Genmab, GSK, ITM, Karyopharm, MSD, Novocure, Oncoinvent, Sanofi, Regeneron, and Seagen, and has participated in consulting data monitoring committees for Abbvie, Agenus, AstraZeneca, Corcept, Daiichi, F. Hoffmann-La Roche, Immunogen, Kronos Bio, Mersana, Novartis, OncXerna, Verastem Oncology, and Zentalis. The remaining authors declare no competing interests.

## Additional information

Tibor A. Zwimpfer[1,2,3]✉, Sian Fereday[1,4], Ahwan Pandey[1], Dinuka Ariyaratne[1], Madawa W. Jayawardana[1,4], Laura Twomey[1], Céline M. Laumont[5], Catherine J. Kennedy[6,7,8], Adelyn Bolithon[9,10], Nicola S. Meagher[9,11], Katy Milne[5], Phineas Hamilton[5], Jennifer Alsop[12], Antonis C. Antoniou[13], George Au-Yeung[1,4], Matthias W. Beckmann[14], Amy Berrington de Gonzalez[15], Christiani Bisinotto[16], Freya Blome[17], Clara Bodelon[18], Jessica Boros[6,7,8], Alison H. Brand[7,8], Michael E. Carney[19], Alicia Cazorla-Jiménez[20], Derek S. Chiu[21], Elizabeth L. Christie[1,4], Anita Chudecka-Głaz[22], Penny Coulson[15], Kara L. Cushing-Haugen[23], Cezary Cybulski[24], Kathleen M. Darcy[25,26], Cath David[27], Trent Davidson[28,29], Arif B. Ekici[30], Esther Elishaev[31], Julius Emons[14], Tobias Engler[32], Rhonda Farrell[8,33], Anna Fischer[17], Montserrat García-Closas[15], Aleksandra Gentry-Maharaj[34,35], Prafull Ghatage[36], Rosalind Glasspool[37], Philipp Harter[38], Andreas D. Hartkopf[32,39], Arndt Hartmann[40], Sebastian Heikaus[41], Brenda Y. Hernandez[42], Anusha Hettiaratchi[43], Sabine Heublein[44], David G. Huntsman[45,46], Mercedes Jimenez-Linan[47], Michael E. Jones[15], Eunyoung Kang[48], Ewa Kaznowska[49], Tomasz Kluz[50], Felix K. F. Kommoss[51], Gottfried Konecny[52], Roy F. P. M. Kruitwagen[53,54], Jessica Kwon[45], Diether Lambrechts[55,56], Cheng-Han Lee[57], Jenny Lester[52], Samuel C. Y. Leung[21], Yee Leung[58,59], Anna Linder[60], Jolanta Lissowska[61], Liselore Loverix[62], Jan Lubiński[63], Constantina Mateoiu[64], Iain A. McNeish[65,66], Malak Moubarak[38], Gregg S. Nelson[36], Nikilyn Nevins[6,7,8], Alexander B. Olawaiye[67], Siel Olbrecht[62], Sandra Orsulic[52], Ana Osorio[68,69], Carmel M. Quinn[43], Ganendra Raj Mohan[58,70], Isabelle Ray-Coquard[71], Cristina Rodríguez-Antona[69,72], Patricia Roxburgh[66], Matthias Ruebner[14], Stuart G. Salfinger[70], Spinder Samra[6,8,73], Minouk J. Schoemaker[15], Hans-Peter Sinn[51], Gabe S. Sonke[74], Linda Steele[75], Colin J. R. Stewart[59], Aline Talhouk[21,45], Adeline Tan[59,76], Christopher M. Tarney[25], Sarah E. Taylor[67], Koen K. Van de Vijver[77,78], Maaike A. van der Aa[79], Toon Van Gorp[80], Els Van Nieuwenhuysen[62], Lilian Van-Wagensveld[53,54,79], Andrea E. Wahner-Hendrickson[81], Christina Walter[32], Chen Wang[82], Julia Welz[38], Nicolas Wentzensen[83], Lynne R. Wilkens[42], Stacey J. Winham[82], Boris Winterhoff[84], Michael S. Anglesio[21,45], Andrew Berchuck[85], Francisco J. Candido dos Reis[16], Paul A. Cohen[58,59], Thomas P. Conrads[25,86], Philip Crowe[9], Jennifer A. Doherty[87], Peter A. Fasching[14], Renée T. Fortner[88,89], María J. García[90], Simon A. Gayther[91], Marc T. Goodman[92], Jacek Gronwald[63], Holly R. Harris[23,93], Florian Heitz[38,41,94], Hugo M. Horlings[95], Beth Y. Karlan[52], Linda E. Kelemen[96], G. Larry Maxwell[25,86], Usha Menon[34], Francesmary Modugno[67,97,98], Susan L. Neuhausen[75], Joellen M. Schildkraut[99], Annette Staebler[17], Karin Sundfeldt[60], Anthony J. Swerdlow[15,100], Ignace Vergote[62], Anna H. Wu[101], James D. Brenton[102], Paul D. P. Pharoah[103], Celeste Leigh Pearce[104], Malcolm C. Pike[101,105], Ellen L. Goode[106], Susan J. Ramus[9,10], Martin Köbel[107], Brad H. Nelson[5,108,109], Anna DeFazio[6,7,8,11], Michael L. Friedlander[27,110,111], David D. L. Bowtell[1,4] & Dale W. Garsed[1,4]✉

[1]Peter MacCallum Cancer Centre, Melbourne, VIC, Australia. [2]Department of Biomedicine, University of Basel, Basel, Switzerland. [3]Gynecological Cancer Centre, University Hospital Basel, Basel, Switzerland. [4]The Sir Peter MacCallum Department of Oncology, The University of Melbourne, Melbourne, VIC, Australia. [5]Deeley Research Centre, BC Cancer, Victoria, BC, Canada. [6]Centre for Cancer Research, The Westmead Institute for Medical Research, Sydney, NSW, Australia. [7]Department of Gynaecological Oncology, Westmead Hospital, Sydney, NSW, Australia. [8]Faculty of Medicine and Health, The University of Sydney, Sydney, NSW, Australia. [9]School of Clinical Medicine, UNSW Medicine and Health, University of NSW Sydney, Sydney, NSW, Australia. [10]Adult Cancer Program, Lowy Cancer Research Centre, University of NSW Sydney, Sydney, New South Wales 2052, Australia. [11]The Daffodil Centre, The University of Sydney, a joint venture with Cancer Council NSW, Sydney, NSW, Australia. [12]Department of Oncology, University of Cambridge, Cambridge, UK. [13]Department of Public Health and Primary Care, Centre for Cancer Genetic Epidemiology, University of Cambridge, Cambridge, UK. [14]Department of Gynecology and Obstetrics, Comprehensive Cancer Center Erlangen-EMN, Friedrich-Alexander University Erlangen-Nuremberg. University Hospital Erlangen, Erlangen, Germany. [15]Division of Genetics and Epidemiology, The Institute of Cancer Research, London, UK. [16]Department of Gynecology and Obstetrics,

Ribeirão Preto Medical School, University of São Paulo, Ribeirão Preto, Brazil. [17]Institute of Pathology and Neuropathology, Tuebingen University Hospital, Tuebingen, Germany. [18]Department of Population Science, American Cancer Society, Atlanta, GA, USA. [19]Department of Obstetrics and Gynecology, John A. Burns School of Medicine, University of Hawaii, Honolulu, HI, USA. [20]Pathology Department, Fundación Jiménez Díaz, Madrid, Spain. [21]British Columbia's Gynecological Cancer Research Team OVCARE, University of British Columbia, BC Cancer and Vancouver General Hospital, Vancouver, BC, Canada. [22]Department of Gynecological Surgery and Gynecological Oncology of Adults and Adolescents, Pomeranian Medical University, Szczecin, Poland. [23]Division of Public Health Sciences, Program in Epidemiology, Fred Hutchinson Cancer Center, Seattle, WA, USA. [24]Department of Comparative Biomedical Sciences, College of Veterinary Medicine, Mississippi State University, Starkville, MS, USA. [25]Department of Gynecologic Surgery and Obstetrics, Gynecologic Cancer Center of Excellence, Uniformed Services University of the Health Sciences, Walter Reed National Military Medical Center, Bethesda, MD, USA. [26]IHenry M Jackson Foundation for the Advancement of Military Medicine, nc, Bethesda, MD, USA. [27]Gynaecological Cancer Centre, Royal Hospital for Women, Randwick, NSW, Australia. [28]NSW Health Pathology, Prince of Wales Hospital, Sydney, NSW, Australia. [29]School of Medicine, Western Sydney University, Penrith, NSW, Australia. [30]Institute of Human Genetics. Comprehensive Cancer Center Erlangen-EMN, University Hospital Erlangen, Friedrich-Alexander University Erlangen-Nuremberg FAU, Erlangen, Germany. [31]Department of Pathology, University of Pittsburgh School of Medicine, Pittsburgh, PA, USA. [32]Department of Women's Health, Tuebingen University Hospital, Tuebingen, Germany. [33]Prince of Wales Private Hospital, Randwick, NSW, Australia. [34]MRC Clinical Trials Unit, Institute of Clinical Trials and Methodology, University College London, London, UK. [35]Department of Women's Cancer, Elizabeth Garrett Anderson Institute for Women's Health, University College London, London, UK. [36]Division of Gynecologic Oncology, Department of Oncology, Cumming School of Medicine, University of Calgary, Calgary, AB, Canada. [37]Beatson West of Scotland Cancer Centre and School of Cancer Sciences, University of Glasgow, Glasgow, UK. [38]Department of Gynecology and Gynecologic Oncology, Evangelische Kliniken Essen-Mitte, Essen, Germany. [39]Department of Gynecology and Obstetrics, University Hospital of Ulm, Ulm, Germany. [40]Institute of Pathology, Comprehensive Cancer Center Erlangen-EMN, Friedrich-Alexander University Erlangen-Nuremberg, University Hospital Erlangen, Erlangen, Germany. [41]Center for Pathology, Evangelische Kliniken Essen-Mitte, Essen, Germany. [42]University of Hawaii Cancer Center, Honolulu, HI, USA. [43]The Health Precincts Biobank, UNSW Biospecimen Services, Mark Wainwright Analytical Centre, UNSW, Sydney, NSW, Australia. [44]Department of Obstetrics and Gynecology, University Hospital Heidelberg, Heidelberg, Germany. [45]Department of Obstetrics and Gynecology, University of British Columbia, Vancouver, BC, Canada. [46]Department of Molecular Oncology, BC Cancer Research Centre, Vancouver, BC, Canada. [47]Department of Histopathology, Addenbrooke's Hospital, Cambridge, UK. [48]Department of Surgery, Seoul National University Bundang Hospital, Seongnam, Republic of Korea. [49]Department of Pathology, Institute of Medical Sciences, Medical College of Rzeszow University, Rzeszow, Poland. [50]Department of Gynecology, Gynecology Oncology and Obstetrics, Institute of Medical Sciences, Medical College of Rzeszów University, Rzeszów, Poland. [51]Institute of Pathology, Heidelberg University Hospital, Heidelberg, Germany. [52]Department of Obstetrics and Gynecology, David Geffen School of Medicine, University of California at Los Angeles, Los Angeles, CA, USA. [53]Department of Obstetrics and Gynecology, Maastricht University Medical Centre, Maastricht, the Netherlands. [54]GROW Ð School for Oncology and Reproduction, Maastricht University Medical Center, Maastricht, the Netherlands. [55]Laboratory for Translational Genetics, Department of Human Genetics, KU Leuven, Leuven, Belgium. [56]VIB Center for Cancer Biology, VIB Leuven, Belgium. [57]Department of Pathology and Laboratory Medicine, University of Alberta, Edmonton, AB, Canada. [58]Department of Gynaecological Oncology, King Edward Memorial Hospital, Subiaco, WA, Australia. [59]Division of Obstetrics and Gynaecology, Medical School, University of Western Australia, Crawley, WA, Australia. [60]Department of Obstetrics and Gynecology, Institute of Clinical Science, Sahlgrenska Center for Cancer Research, University of Gothenburg, Gothenburg, Sweden. [61]Department of Cancer Epidemiology and Prevention, M Sklodowska-Curie National Research Oncology Institute, Warsaw, Poland. [62]Division of Gynecologic Oncology, Department of Gynecology and Obstetrics, Leuven Cancer Institute, Leuven, Belgium. [63]Department of Genetics and Pathology, Pomeranian Medical University, Szczecin, Poland. [64]Department of Pathology, University of Gothenburg, Gothenburg, Sweden. [65]Division of Cancer and Ovarian Cancer Action Research Centre, Department Surgery & Cancer, Imperial College London, London, UK. [66]School of Cancer Sciences, University of Glasgow, Glasgow, UK. [67]Division of Gynecologic Oncology, Department of Obstetrics, Gynecology and Reproductive Sciences, University of Pittsburgh School of Medicine, Pittsburgh, PA, USA. [68]Genetics Service, Fundación Jiménez Díaz University Hospital and Health Research Institute, Universidad Autónoma de Madrid IIS-FJD, UAM, Madrid, Spain. [69]Centre for Biomedical Network Research on Rare Diseases CIBERER, Instituto de Salud Carlos III, Madrid, Spain. [70]Department of Gynaecological Oncology, St John of God Subiaco Hospital, Subiaco, WA, Australia. [71]Centre Leon Berard and University Claude Bernard Lyon 1, Lyon, France. [72]Pharmacogenomics and Tumor Biomarkers Group, Institute for Biomedical Research Sols-Morreale CSIC-UAM, Madrid, Spain. [73]Tissue Pathology and Diagnostic Oncology, Westmead Hospital, Sydney, NSW, Australia. [74]Department of Medical Oncology, Netherlands Cancer Institute, Amsterdam, The Netherlands. [75]Department of Population Sciences, Beckman Research Institute of City of Hope, Duarte, CA, USA. [76]Gynaepath WA, Clinipath Sonic Healthcare, Osbourne Park, Australia. [77]Department of Pathology, Ghent University Hospital, Cancer Research Institute Ghent CRIG, Ghent, Belgium. [78]Department of Pathology, Antwerp University Hospital, Antwerp, Belgium. [79]Department of Research, Netherlands Comprehensive Cancer Organization IKNL, Utrecht, the Netherlands. [80]Division of Gynaecological Oncology, Leuven Cancer Institute, University Hospital Leuven and KU Leuven, Leuven, Belgium. [81]Department of Oncology, Mayo Clinic, Rochester, MN, USA. [82]Department of Quantitative Health Sciences, Division of Computational Biology, Mayo Clinic, Rochester, MN, USA. [83]Division of Cancer Epidemiology and Genetics, National Cancer Institute, Bethesda, MD, USA. [84]Department of Obstetrics, Gynecology and Women's Health, University of Minnesota, Minneapolis, MN, USA. [85]Division of Gynecologic Oncology, Department of Obstetrics and Gynecology, Duke University Medical Center, Durham, NC, USA. [86]Women's Health Integrated Research Center, Women's Service Line, Inova Health System, Falls Church, VA, USA. [87]Huntsman Cancer Institute, Department of Population Health Sciences, University of Utah, Salt Lake City, UT, USA. [88]Division of Cancer Epidemiology, German Cancer Research Center DKFZ, Heidelberg, Germany. [89]Department of Research, Cancer Registry of Norway, Norwegian Institute of Public Health, Oslo, Norway. [90]Genomic Biomarkers and Precision Oncology Group, Sols-Morreale Biomedical Research Institute IIBM, Consejo Superior de Investigaciones Cientficas & Universidad Autónoma de Madrid CSIC-UAM, Madrid 28029, Spain. [91]Medicine, University of Texas Health, San Antonio, TX, USA. [92]Cancer Prevention and Control Program, Cedars-Sinai Cancer, Cedars-Sinai Medical Center, Los Angeles, CA, USA. [93]Department of Epidemiology, University of Washington School of Public Health, Seattle, WA, USA. [94]Department of Gynecology and Gynecological Oncology, HSK, Dr Horst-Schmidt Klinik, Wiesbaden, Wiesbaden, Germany. [95]Department of Pathology, The Netherlands Cancer Institute - Antoni van Leeuwenhoek hospital, Amsterdam, the Netherlands. [96]Communicable Disease Epidemiology Section, South Carolina Department of Public Health, Columbia, SC, USA. [97]Department of Epidemiology, University of Pittsburgh School of Public Health, Pittsburgh, PA, USA. [98]Women's Cancer Research Center, Magee-Womens Research Institute and Hillman Cancer Center, Pittsburgh, PA, USA. [99]Department of Epidemiology, Rollins School of Public Health, Emory University, Atlanta, GA, USA. [100]Division of Breast Cancer Research, The Institute of Cancer Research, London, UK. [101]Department of Population Health and Public Health Sciences, Keck School of Medicine, University of Southern California Norris Comprehensive Cancer Center, Los Angeles, CA, USA. [102]Cancer Research UK Cambridge Institute, University of Cambridge, Cambridge, UK. [103]Department of Computational Biomedicine, Cedars-Sinai Medical Center, West Hollywood, CA, USA. [104]Department of Epidemiology, University of Michigan School of Public Health, Ann Arbor, MI, USA. [105]Department of Epidemiology and Biostatistics, Memorial Sloan-Kettering Cancer Center, New York, NY, USA. [106]Division

of Epidemiology, Department of Quantitative Health Sciences, Mayo Clinic, Rochester, MN, USA. [107]Department of Pathology and Laboratory Medicine, Foothills Medical Center, University of Calgary, Calgary, AB, Canada. [108]Department of Biochemistry and Microbiology, University of Victoria, Victoria, BC, Canada. [109]Department of Medical Genetics, University of British Columbia, Victoria, BC, Canada. [110]Nelune Comprehensive Cancer Centre, Prince of Wales Hospital, Sydney, NSW, Australia. [111]Prince of Wales Clinical School, UNSW Medicine and Health, University of NSW Sydney, Sydney, NSW, Australia. ✉e-mail: tibor.zwimpfer@petermac.org; Dale.Garsed@petermac.org

