## [Transparent Peer Review file · Nature Communications]

Clinicopathologic and molecular predictors in BRCA-deficient tubo-ovarian high-grade serous carcinoma

Corresponding Author: Dr Dale Garsed

Version 0:

Reviewer comments:

Reviewer #1

(Remarks to the Author)

This manuscript addresses an important question, which is why some patients with BRCA1/2 mutant high-grade serous ovarian carcinoma (HGSC), have better outcomes than others.

Authors profiled 154 tumors, enriched for patients with BRCA-deficient tumors that experienced short overall survival (≤ 3 years, $n=42$), using whole-genome, transcriptome, and methylation analyses and then followed this up by analysis of a very large HGSC cohort, finding that:

- Association of residual disease with prognosis is attenuated in gBRCApv-carriers
- gBRCApv location and type are associated with survival and therapy response
- NF1 gene alterations are associated with improved survival in BRCA2-deficient HGSC
- PIK3CA and RAD21 amplifications are associated with short survival in BRCA2-deficient HGSC
- Elevated HRD scarring is prognostic for survival in BRCA-deficient HGSC
- CD8+ PD-1+ T cells are prognostic for survival in gBRCApv-carriers
- The mesenchymal features c-KIT and mast cells are associated with poor outcome in HGSC

The major strengths of this work are the cohort size and its diversity (over 7,000 patients across multiple cohorts (AOCS, MOCOG, OTTA) provides a comprehensive analysis.

The data described could be widely used by the community and so reporting these observations is important.

There are some suggestions that could improve the clarity of the work provided below;

Major points:

1. The cohort spans nearly two decades, during which treatment paradigms (e.g., PARPi availability) evolved. This may confound survival analyses and should be more explicitly addressed – for example sub analysis of those who only received platinum and not PARPi might be warranted if numerically possible?

2. The authors claim that HRD is not a binary property but rather there are additional modifiers of outcomes. This is not new observation nor concept. Since early studies (e.g. PMID: 26957554) introduced the use of binary HRD scores, there have been many studies suggesting that HRD is a context-dependent phenomenon, and not a simply binary phenotype (PMID: 39073402). HRD score should be interpreted in the specific biological and clinical context.

3. The concept / observation that BRCA mutation location stratifies outcome is not novel e.g. PMID: 36564284, PMID: 40174507. In addition, how the authors design the analysis that leads to this conclusion deserves further thought: in the manuscript presented, authors predominantly compare outcomes in BRCA1/2 mutation carriers with mutations in different positions in the genes to outcomes in HGSC patients with no BRCA1/2 mutation, as opposed to comparing outcomes in cohorts of patients with BRCA1/2 mutations in certain positions with other BRCA1/2 mutant patients with mutations in different positions. Conversely when different BRCA mutation types are correlated with outcome (e.g. frameshift, missense etc), outcome for those with one mutation type is compared to outcome with other mutation types. I think for clarity and logic, both forms of analysis should be shown and conclusions drawn appropriately.

4. Some subgroup comparisons (e.g., BRCA1 exon 10 Δ 11q expression) lack statistical significance due to small sample sizes. This limits interpretability and generalizability.
5. The authors found that the BRCA1- Δ 11q splice isoform is expressed across many tumors, regardless of BRCA mutation status, indicating that it represents a common transcript variant. Interestingly, BRCA1 carriers in exon 10 showed significantly higher Δ 11q expression ($P = 0.011$), suggesting that these tumors may use this isoform as a compensatory mechanism to bypass the truncated exon. In fact, the authors show that patients with high Δ 11q isoform expression had shorter overall survival (median = 2.74 years), even though they maintain high HRD scores. This is an interesting observation that extends prior work, but not a novel finding though (PMID: 27197267).
6. The overall robustness of the analysis could be improved by applying other measures of HRD to the tumor cohort other than CHORD.
7. The authors indicate in the section “Association of residual disease with prognosis is attenuated in gBRCApv-carrier” that the proportional hazards assumption did not hold, or was violated, when applying the survival model, thus they used an Accelerated Failure Time (AFT) model to estimate survival differences between groups, therefore considering that time-dependent effects exist. It clearly suggests that the survival advantage for BRCA carriers is time-limited. In fact, this effect is observed in both studied cohorts in Figure 1a and 1b, where the biggest drop in the survival curve for gBRCApv RO patients occur between year 2 and year 3 (highest number of patients died). Could please the authors explore in detail this effect and specify if there is a therapeutic time window identified? Showing the survival curves for the progression free survival would help clarify this.
8. In addition, it is well documented that the presence of reversion mutations is directly associated with therapeutic response. The cohorts AOCS and OTTA are long-term cohorts of patients treated with standard of care at the time of diagnosis (which would have included platinum, but also more recently PARP inhibitor). Could the authors please comment on whether biopsies exist to assess the presence of reversion mutations (and if so please assess reversions in these).
9. Authros conclude that “BRCA1 and BRCA2 domains associated with prolonged survival were more likely to have truncating variants than missense or splice site variants”. Have the authors taken into consideration the possibility that this correlation is driven by some domains generally lacking missense and splice mutations (i.e. it is not the type of mutation that is the driver of this effect but the position)?
10. Some of the conclusions, where not novel, are not always annotated as such (this is a repetitive issue) e.g. “CCNE1 amplifications (gene level copy number ≥ 7) were associated with BRCA-proficiency, and particularly the short-survival BRCA-proficient group” and “BRCA-proficient tumors had less genomic scarring and were associated with an older age at diagnosis compared to BRCA1-deficient and BRCA2-deficient tumors”. A casual reader might think these are novel observations.
11. The conclusions might be better substantiated if some of the key observations were functionally investigated. For example, the combination of BRCA2-deficiency and loss-of-function NF1 alteration was associated with the best survival outcome – an obvious experiment might be to see if NF1 loss of function alters platinum or PARP inhibitor response in BRCA2 mutant cells. Similar experiments could be conducted on PIK3Ca and RAD21.
12. There is a recurrent lack of clarity across the vast majority of the figures (main and supplementary), regarding which dataset is used for each analysis. Please, could the authors directly indicate within each figure this information? It will help the readers to go through the study easily, given the number of different datasets used for different purposes.
13. In Supplementary Table S2, the authors disclose the mutations studied in the AOCS cohort. Since the gBRCApv-carriers OTTA cohort is also used to validate some of the findings, would it be possible for the authors to provide the list of mutations identified in the OTTA cohort as well please?
14. In addition, since the distribution and frequency of mutations are depicted in the Extended Data Figure 4B.A, could the authors please make a comment on whether any of the observed results might be influenced by the presence of founder mutations? If it is possible, in both AOCS and OTTA cohort please.

Reviewer #2

(Remarks to the Author)

This is an in-depth and very interesting study of genomic features of BRCA1 and BRCA2 mutated ovarian cancer that impact survival in ovarian cancer patients. This study has a number of important findings to direct future clinical trials.

Major comments:

1. A major issue is that the discussion seems unfinished. There is not a limitations section nor a concluding paragraph. The Discussion simply ends abruptly after discussing NF1 loss and PIK3CA amplifications, with no discussion of RAD21 amplifications. More importantly there is no discussion of what the authors want us to take home from their study and how we should use this data to direct prospective clinical trials and/or tailor treatment.
2. The abstract is out of order from the Figures which decreases readability as the reader is setup to see the HRD and mutational data (Fig.4 and Fig.3, respectively) first and then the mutation location data (Fig. 2) and then the residual disease data (Fig. 1). Suggest considering to reorder the abstract.
3. The conclusion sentence of the abstract stating that "HR proficiency is not a binary property" is unclear to me – what is the take-home message the authors are trying to portray?
4. In Figure 3, is there any difference in TP53 mutation status between BRCA1 LTS/STS, BRCA2 LTS/STS and BRCA-P tumors? While I understand that most HGPSC have TP53 mutations, some do not.
5. RAD21 amplification is often seen with MYC amplification, is this the case here? Are the survival differences in BRCA2 attributable to both MYC and RAD21 amplification?
6. The point of Figure 3 and Extended Data Fig.7-8 is to show that PIK3CA amp, RAD21 amp, and absence of NF1 loss are associated with STS in BRCA2 only; however, the box plots need significance testing to point out that difference, in addition the KM curves are a better demonstration of the effect.
7. In Figure 5, it appears that activated dendritic cells are very high in IMMB.5 which has the worst survival but in Extended Data Figure 9, there does not appear to be an effect of activated dendritic cells on survival; however, it is stated in the text that mast cells are the cell type most strongly associated with short survival – it is curious as the heatmap does not seem to support this conclusion.
8. In the discussion, might be worthwhile to point out that exon 11 skipping mutations are also associated with a reduced penetrance cancer risk phenotype, further supporting that these mutations are less "BRCA-like"
9. The association of immune phenotypes and survival in BRCA deficient ovarian cancer has also been described in PMID: 35201851, suggest discussing a comparison to this study.

Minor comments:

- 1) Mutation location analysis in abstract is not very clear
- 2) Table S1/S2 – what are the "publication IDs" referring to?
- 3) Figure 1a/b – consider placing label of cohort on each graph for easier readability
- 4) Extended Data Fig 2a – there is no data for non-carriers & R0
- 5) Fig.3a – the HRD score colors between Low and Moderate are hard to distinguish when printed (although a bit easier to discern on the computer screen)
- 6) Fig.4f – the colors of the KM curves are hard to distinguish when printed (although a bit easier to discern on the computer screen)
- 7) Extended Data Fig.4 is title "Prognostic significant of pathogenic germline BRA1 and BRCA2 variant by domain location and mutation type on outcomes in HGSC" but there is no data on prognosis or outcomes, it is just a distribution of variant types.
- 8) Extended Data Fig.9b – what is this showing the expression of? There is no label

Version 1:

Reviewer comments:

Reviewer #1

(Remarks to the Author)

In their point by point response the authors have made all reasonable attempts to either clarify prior analyses and/or explain limitations of their study. With this in mind I have no further comments to make other than I feel their manuscript would make a useful addition to the literature.

Reviewer #2

(Remarks to the Author)

The authors did not respond to my Major comment 7: In Figure 5, it appears that activated dendritic cells are very high in IMMB.5 which has the worst survival but in Extended Data Figure 9, there does not appear to be an effect of activated dendritic cells on survival; however, it is stated in the text that mast cells are the cell type most strongly associated with short survival – it is curious as the heatmap does not seem to support this conclusion. The authors explained a response but did not edit the text. I suggest adding that the association with dendritic cells did not remain significant in the multivariable Cox model on page 24.

The authors responded well to all other reviewer comments.

Reviewer #1

Major comment 1:

The cohort spans nearly two decades, during which treatment paradigms (e.g., PARPi availability) evolved. This may confound survival analyses and should be more explicitly addressed – for example subanalysis of those who only received platinum and not PARPi might be warranted if numerically possible.

Response:

We agree that evolving treatment paradigms may influence survival outcomes. PARP inhibitor use in the first-line setting was already included as an adjustment factor in our original multivariable analysis to account for potential confounding. In addition, as only 39 out of 1,389 patients received PARP inhibitor maintenance therapy, we performed subgroup analyses restricted to patients who only received platinum-based chemotherapy. We found that the survival associations persisted within this subgroup, suggesting that our main conclusions are not driven by exposure to PARP inhibitors. These results are presented in the supplementary Table S5 and described on page 14, lines 353-354 of the revised results section.

Major comment 2:

The authors claim that HRD is not a binary property but rather there are additional modifiers of outcomes. This is not new observation nor concept. Since early studies (e.g. PMID: 26957554) introduced the use of binary HRD scores, there have been many studies suggesting that HRD is a context-dependent phenomenon, and not a simply binary phenotype (PMID: 39073402). HRD score should be interpreted in the specific biological and clinical context.

Response:

We have revised the text (Abstract, page 12, line 272; Introduction, page 12, line 294; and Discussion, page 25, line 700) to better reflect that the concept of HRD as a context-dependent rather than binary phenomenon has been previously reported. We have also added relevant references (<https://doi.org/10.1038/s41467-020-19406-4>; <https://doi.org/10.17305/bb.2024.10448>), in addition to those already included (e.g., <https://www.nature.com/articles/s41598-020-59671-3>), to place our findings in the context of previous studies.

Major comment 3:

The concept / observation that BRCA mutation location stratifies outcome is not novel e.g. PMID: 36564284, PMID: 40174507. In addition, how the authors design the analysis that leads to this conclusion deserves further thought: in the manuscript presented, authors predominantly compare outcomes in BRCA1/2 mutation carriers with mutations in different positions in the genes to outcomes in HGSC patients with no BRCA1/2 mutation, as opposed to comparing outcomes in cohorts of patients with BRCA1/2 mutations in certain positions with other BRCA1/2 mutant patients with mutations in different positions. Conversely when different BRCA mutation types are correlated with outcome (e.g. frameshift, missense etc), outcome for those with one mutation type is compared to outcome with other mutation types. I think for clarity and logic, both forms of analysis should be shown and conclusions drawn appropriately.

Response:

We agree that mutation location and type as prognostic and predictive factors have been

previously described in the context of PARPi, and we have cited the relevant literature accordingly (Results, pages 15 and 17, lines 383 and 432; Discussion, page 26, line 727). We also thank the reviewer for highlighting the recent publication by Marchetti et al. (PMID: 40174507), which we had not included previously, this reference has now been added. Furthermore, to address the reviewer's comment, we have incorporated the results of the survival analysis by mutation type into Table 3 and described these findings in the revised Results section (page 17, lines 437-442).

Major comment 4:

Some subgroup comparisons (e.g., BRCA1 exon 10 z111q expression) lack statistical significance due to small sample sizes.

Response:

We acknowledge this limitation and have clarified it explicitly in the Discussion (page 27, lines 735-738).

Major comment 5:

The authors found that the BRCA1-z111q splice isoform is expressed across many tumors, regardless of BRCA mutation status, indicating that it represents a common transcript variant. Interestingly, BRCA1 carriers in exon 10 showed significantly higher z111q expression ($P = 0.011$), suggesting that these tumors may use this isoform as a compensatory mechanism to bypass the truncated exon. In fact, the authors show that patients with high z111q isoform expression had shorter overall survival (median = 2.74 years), even though they maintain high HRD scores. This is an interesting observation that extends prior work, but not a novel finding though (PMID: 27197267).

Response:

We acknowledge that our analysis builds upon and extends prior work on the BRCA1- $\Delta 11q$ isoform. Accordingly, we have cited the relevant studies (<https://doi.org/10.1158/0008-5472.CAN-16-0186>, <https://pubmed.ncbi.nlm.nih.gov/39103848/>) in the revised Results section (page 16, line 411) and Discussion (pages 27, lines 734 and 738-744).

Major comment 6:

The overall robustness of the analysis could be improved by applying other measures of HRD to the tumor cohort other than CHORD.

Response:

The analysis of the tumor cohort includes consideration of 1) germline and somatic alterations of *BRCA* and non-*BRCA* HR genes, 2) copy number based HRD scores (sum of large-scale transitions, loss-of-heterozygosity and telomeric allelic imbalance), as well as 3) CHORD, which incorporates multiple types of mutational signatures associated with HRD. Following this suggestion, we have now also applied HRDetect (<https://doi.org/10.1038/nm.4292>) to the tumor cohort to further evaluate HRD status and assess *BRCA*-deficiency assignments.

HRDetect scores showed excellent concordance with both the CHORD classifier and our pathogenic HR gene variant curation: using the HRDetect cutoff of >0.7

(<https://doi.org/10.1038/s41467-020-19406-4>), 100% of *BRCA1*-deficient tumors (71/71) and 94% of *BRCA2*-deficient tumors (31/33) were classified as HRDetect-high, whereas only 6% of *BRCA*-proficient tumors (3/49) exceeded this threshold. These results demonstrate strong agreement between HRDetect, CHORD, and HRD scores, supporting the robustness of our HRD assignments. We have added a description of this complementary HRDetect analysis to the Supplementary Information (Page 4, lines 110-115) and included the corresponding results in Supplementary Table S11.

Major comment 7:

The authors indicate in the section “Association of residual disease with prognosis is attenuated in gBRCApv-carrier” that the proportional hazards assumption did not hold, or was violated, when applying the survival model, thus they used an Accelerated Failure Time (AFT) model to estimate survival differences between groups, therefore considering that time-dependent effects exist. It clearly suggests that the survival advantage for BRCA carriers is time-limited. In fact, this effect is observed in both studied cohorts in Figure 1a and 1b, where the biggest drop in the survival curve for gBRCApv RO patients occur between year 2 and year 3 (highest number of patients died). Could please the authors explore in detail this effect and specify if there is a therapeutic time window identified? Showing the survival curves for the progression free survival would help clarify this.

Response:

As noted by the reviewer, the proportional hazards assumption was violated, indicating the presence of time-dependent effects and supporting the use of an AFT model. To characterize this further, we examined the shape of both OS and PFS curves. In Figures 1a, b, the steepest decline among *gBRCA*pv-carriers with no residual disease occurs between years 2-3. As suggested, we have now added the corresponding PFS curves from the AOCS cohort (Extended Data Fig. 2b), which similarly show early separation between *gBRCA*pv-carriers and non-carriers that begins to narrow after approximately two years. PFS data for the OTTA cohort were not available.

This pattern is consistent with what is observed for OS and suggests that the survival advantage in *gBRCA*pv-carriers is predominantly driven by enhanced initial platinum responsiveness rather than by sustained long-term resistance prevention. The manuscript has been updated accordingly (page 14, lines 353-360).

Major comment 8:

In addition, it is well documented that the presence of reversion mutations is directly associated with therapeutic response. The cohorts AOCS and OTTA are long-term cohorts of patients treated with standard of care at the time of diagnosis (which would have included platinum, but also more recently PARP inhibitor). Could the authors please comment on whether biopsies exist to assess the presence of reversion mutations (and if so please assess reversions in these).

Response:

We agree that assessing reversion mutations would provide valuable insights into treatment response and resistance mechanisms. However, tumor biopsies suitable for this type of analysis are not available for the OTTA cohort, and the limited relapse samples available from the AOCS do not overlap with the patients in our cohort. Consequently, we were unable to assess reversion mutations within the current study. However, we note this in the Discussion and refer readers to previous work that has explored *BRCA* reversions and other acquired resistance mechanisms in ovarian cancer (page 25, lines 688-692; page 27, lines 738-744).

Major comment 9:

Authors conclude that “BRCA1 and BRCA2 domains associated with prolonged survival

were more likely to have truncating variants than missense or splice site variants”.
Have the authors taken into consideration the possibility that this correlation is driven by some domains generally lacking missense and splice mutations (i.e. it is not the type of mutation that is the driver of this effect but the position)?

Response:

It is correct that the apparent association between mutation type and survival could be influenced by positional clustering of variants within specific BRCA1 or BRCA2 domains, as shown in Extended Data Fig. 4a-d. While our sample sizes did not allow for simultaneous adjustment for both mutation type and domain due to limited numbers within each category, we specifically examined survival according to variant position (exon and functional domain) and, following your suggestion (major comment 3), have now also incorporated mutation type into the adjusted AFT models (Table 3).

These analyses, adjusted for clinical covariates including FIGO stage, residual disease, age, and treatment factors, demonstrate that variant location itself is associated with overall survival. Thus, even without jointly modelling mutation type, the positional analyses indicate that the survival associations are not solely explained by variant class distribution. As described in our response to Major Comment 3, we have expanded the Results section (page 17, lines 437-442)

Major comment 10:

Some of the conclusions, where not novel, are not always annotated as such (this is a repetitive issue) e.g. “CCNE1 amplifications (gene level copy number ≥ 7) were associated with BRCA-proficiency, and particularly the short-survival BRCA-proficient group” and “BRCA-proficient tumors had less genomic scarring and were associated with an older age at diagnosis compared to BRCA1-deficient and BRCA2-deficient tumors”. A casual reader might think these are novel observations.

Response:

We have revised the Results section to explicitly acknowledge findings consistent with prior studies (page 18, lines 457 and 459-460) and have added the relevant references.

Major comment 11:

The conclusions might be better substantiated if some of the key observations were

functionally investigated. For example, the combination of BRCA2-deficiency and loss-of-function NF1 alteration was associated with the best survival outcome – an obvious experiment might be to see if NF1 loss of function alters platinum or PARP inhibitor response in BRCA2 mutant cells. Similar experiments could be conducted on PIK3Ca and RAD21.

Response:

We agree that functional studies exploring the interaction between *BRCA2* deficiency and *NF1*, *PIK3CA*, or *RAD21* alterations would be valuable to better understand the biological mechanisms underlying our observations. While these experiments are beyond the scope and feasibility of the current study, we intend to pursue such investigations in future work to build upon the findings presented here. Where possible, we have tested key observations in independent cohorts/datasets, for example, the association between *NF1* expression and survival in the statistically powered OTTA cohort (n=5666 patients with survival status and *NF1* mRNA data).

Major comment 12:

There is a recurrent lack of clarity across the vast majority of the figures (main and supplementary), regarding which dataset is used for each analysis. Please, could the authors directly indicate within each figure this information? It will help the readers to go through the study easily, given the number of different datasets used for different purposes.

Response:

We have revised all figures and tables to clearly indicate the dataset(s) used (AOCS, OTTA, MOCOG). This information is now explicitly stated in the figure and table legends, or directly within the figures where appropriate.

Major comment 13:

In Supplementary Table S2, the authors disclose the mutations studied in the AOCS cohort. Since the gBRCApv-carriers OTTA cohort is also used to validate some of the findings, would it be possible for the authors to provide the list of mutations identified in the OTTA cohort as well please?

Response:

We appreciate the interest in accessing the detailed variant data from the OTTA cohort.

Unfortunately, individual-level *BRCA* variant information for the OTTA cohort is not available to us for public release under the terms of the consortium's data-sharing agreements, which restrict dissemination of potentially identifiable germline variant data. We have clarified this limitation and provided appropriate wording in the Data Availability section, noting that detailed OTTA variant-level data can be accessed through the OTTA consortium upon request and appropriate data access approval.

Major comment 14:

In addition, since the distribution and frequency of mutations are depicted in the Extended Data Figure 4B.A, could the authors please make a comment on whether any of the observed results might be influenced by the presence of founder mutations? If it is possible, in both AOCS and OTTA cohort please.

Response:

We systematically examined all germline *BRCAJ/2* pathogenic variants in the AOCS cohort for known population founder mutations using available evidence in the literature and ClinVar founder annotations (Supplementary Table S2). Founder variants identified in AOCS were most commonly *BRCAJ* c.68_69del, c.181T>G, c.4035del, c.4065_4068del, and c.5266dup, and *BRCA2* c.2808_2811del, c.3847_3848del, c.658_659del, c.771_775del, and c.5946del. As detailed mutation-level information was not available for OTTA participants, this analysis could not be performed for the OTTA cohort.

Analysis of their distribution in AOCS showed that founder mutations were statistically significant enriched in the *BRCAJ* RING domain, but not in *BRCAJ* BRCT or DBD regions, nor in any *BRCA2* functional domain. Founder status was not associated with mutation type in either gene.

We have added a description of this analysis to the Results (Pages 16 and 17, lines 408-410 and 437-442, Extended Data Figs. 5c,d,g,h, Supplementary Table S2) and Supplementary Information (Page 3, lines 81-97).

Reviewer #2

Major comment 1:

A major issue is that the discussion seems unfinished. There is not a limitations section nor a

concluding paragraph. The Discussion simply ends abruptly after discussing NF1 loss and PIK3CA amplifications, with no discussion of RAD21 amplifications. More importantly there is no discussion of what the authors want us to take home from their study and how we should use this data to direct prospective clinical trials and/or tailor treatment.

Response:

We have revised the Discussion to include dedicated Limitations and Conclusions subsections. These outline sample-size constraints, retrospective design, and translational implications for tailoring clinical trials (page 28 and 29, lines 782-802).

Major comment 2:

The abstract is out of order from the Figures which decreases readability as the reader is setup to see the HRD and mutational data (Fig.4 and Fig.3, respectively) first and then the mutation location data (Fig. 2) and then the residual disease data (Fig. 1). Suggest considering to reorder the abstract.

Response:

We have reordered the Abstract to follow the logical flow of the Figures and the Results text (residual disease → Mutation location → Mutational data → HRD → immune contexture), to enhance readability.

Major comment 3:

The conclusion sentence of the abstract stating that “HR proficiency is not a binary property” is unclear to me – what is the take-home message the authors are trying to portray?

Response:

The concluding sentence of the abstract has been rewritten for clarity.

Major comment 4:

In Figure 3, is there any difference in TP53 mutation status between BRCA1 LTS/STS, BRCA2 LTS/STS and BRCA-P tumors? While I understand that most HGPSC have TP53 mutations, some do not.

Response:

TP53 mutations were present in nearly all tumors (Supplementary Table S13), consistent with HGSC. This information has now also been added to Fig. 3a. Only one tumor (BRCA_1) in the *BRCA2* STS subgroup lacked a pathogenic *TP53* mutation, which was confirmed by p53 IHC. An RB1 loss-of-function mutation was the only notable somatic alteration in this tumor and there was no evidence on pathological review to suggest this case was misclassified as HGSC. Therefore, no differences in *TP53* mutation frequency were observed between *BRCA1* LTS/STS, *BRCA2* LTS/STS, and *BRCA*-P tumors.

In addition, we have previously shown that *TP53* mutation type is not associated with differences in outcomes in HGSC (PMID: 29061645, PMID: 36948887).

Major comment 5:

RAD21 amplification is often seen with *MYC* amplification, is this the case here? Are the survival differences in *BRCA2* attributable to both *MYC* and *RAD21* amplification?

Response:

We thank the reviewer for this important point. In our cohort, *RAD21* amplification frequently co-occurred with *MYC* amplification in *BRCA2*-deficient tumors, as demonstrated by a significant co-occurrence in the mutual exclusivity analysis (23/28, *P*_{adj} < 0.001; Supplementary Table S17). Co-amplification of *RAD21* and *MYC* was observed in 20.6% (8/34) of *BRCA2*-deficient tumors.

Regarding clinical outcome, survival analyses based on copy-number status indicate that the adverse prognosis in *BRCA2*-deficient tumors is attributable to amplification of both *RAD21* and *MYC*. However, at the transcriptomic level, *MYC* copy-number gains showed only a weak correlation with RNA expression, and *MYC*-amplified tumors did not exhibit significantly increased *MYC* expression, whereas *RAD21* amplification was strongly correlated with elevated mRNA expression. Thus, while outcome differences are observed at the genomic amplification level for both genes, these effects are not as strong reflected at the RNA expression level for *MYC*. Together, these findings indicate that while *MYC* and *RAD21* amplifications both contribute to adverse outcome at the genomic level in *BRCA2*-deficient tumors, the prognostic impact of *MYC* is not consistently reflected at the transcriptomic level. To address this comment, we have adapted the Results section accordingly (page 20, lines 518-537 and 562-569, tables S17, S18, S21), and we have added an additional *MYC*-focused

analysis to the Supplementary Information (page 8-10, lines 199-233, Supplementary Fig. 4a,b,c,d) to more comprehensively document these findings.

Major comment 6:

The point of Figure 3 and Extended Data Fig. 7-8 is to show that PIK3CA amp, RAD21 amp, and absence of NF1 loss are associated with STS in BRCA2 only; however, the box plots need significance testing to point out that difference, in addition the KM curves are a better demonstration of the effect.

Response:

We have added the results of the pairwise Fisher's exact tests comparing the frequency of *PIK3CA* amplification, *RAD21* amplification, and *NF1* loss across all *BRCA*-survival subgroups to Supplementary Table S14. In addition, as noted by the reviewer, the Kaplan-Meier curves illustrating the impact of these alterations on survival are already provided in the Extended Data Figures 7c, and 9c,d, with corresponding results summarized in Supplementary Table S11.

Major comment 7:

In Figure 5, it appears that activated dendritic cells are very high in IMMB.5 which has the worst survival but in Extended Data Figure 9, there does not appear to be an effect of activated dendritic cells on survival; however, it is stated in the text that mast cells are the cell type most strongly associated with short survival – it is curious as the heatmap does not seem to support this conclusion.

Response:

The apparent discrepancy arises from differences between the unsupervised clustering results shown in Figure 5 and the multivariable Cox proportional-hazards model in Extended Data Fig. 10a (previously Extended Data Fig. 9a). In the heatmap, the IMMB.5 subtype (n=24) indeed shows elevated activated dendritic-cell scores, reflecting their co-enrichment within an inflammatory, mesenchymal-like microenvironment that also contains high mast-cell signatures. However, when modeled across all immune variables and all patients (n=153) simultaneously, the multivariable Cox analysis identified resting mast cells as the immune population most strongly and independently associated with poor survival, whereas activated dendritic cells did not retain significance after adjustment for correlated immune cell types.

Major comment 8:

In the discussion, might be worthwhile to point out that exon 11 skipping mutations are also associated with a reduced penetrance cancer risk phenotype, further supporting that these mutations are less “BRCA-like”.

Response:

Thank you for this suggestion. While our cohort did not have many of the reduced penetrance pathogenic variants (PMID: 39488595) and we were unable to assess their associated tumor “BRCA-ness”, we have added some text to the Discussion (page 27, lines 738-744) pointing to this literature.

Major comment 9:

The association of immune phenotypes and survival in BRCA deficient ovarian cancer has also been described in PMID: 35201851, suggest discussing a comparison to this study.

Response:

We thank the reviewer for highlighting this important study. We have now incorporated a comparison to Kraya et al. (PMID: 35201851) into the Discussion (pages 27, lines 749-754). We discuss how their identification of immune-high and immune-low subsets within BRCA-deficient ovarian cancer aligns with our observation that CD8+PD-1+ T-cell infiltration is prognostic, while also noting that different tumor-intrinsic features may underlie immune heterogeneity across cohorts.

Minor comment 1:

Mutation location analysis in abstract is not very clear

Response:

The abstract text has been modified to clarify this point (pages 11 and 12, lines 254-274).

Minor comment 2:

Table S1/S2 – what are the “publication IDs” referring to?

Response:

The “Publication IDs” are de-identified participant identifiers generated specifically for use in publications. For clarity, the label of this column in Supp Tables 1 & 2 has been changed to “Participant ID”.

Minor comment 3:

Figure 1a/b – consider placing label of cohort on each graph for easier readability

Response:

We have added clear cohort labels directly to each panel of Figure 1a/b to improve readability and ensure that the corresponding cohort is immediately identifiable.

Minor comment 4:

Extended Data Fig 2a – there is no data for non-carriers & R0

Response:

The non-carriers & R0 group serves as the reference group / baseline for comparison in this analysis, as labelled on the left side of Extended Data Fig. 2a (i.e., Ref = Non carriers & R0).

Minor comment 5:

Fig.3a – the HRD score colors between Low and Moderate are hard to distinguish when printed (although a bit easier to discern on the computer screen)

Response:

We have updated the HRD score classification color scheme in Fig. 3a, as well as the related figures in the manuscript (Fig. 5a, Extended Fig. 6b), using a color scheme that is discernible when printed in color or black and white.

Minor comment 6:

Fig.4f – the colors of the KM curves are hard to distinguish when printed (although a bit easier to discern on the computer screen)

Response:

In the revised figure, we have adjusted the color palette to improve visual distinguishability, including when printed.

Minor comment 7:

Extended Data Fig.4 is title “Prognostic significant of pathogenic germline BRCA1 and BRCA2 variant by domain location and mutation type on outcomes in HGSC” but there is no data on prognosis or outcomes, it is just a distribution of variant types.

Response:

We have revised the title of Extended Data Fig. 5 (previously Extended Data Fig. 4) to accurately reflect its content: *“Distribution of pathogenic germline BRCA1 and BRCA2 variants by domain location and mutation type in HGSC (AOCS cohort)”*.

Minor comment 8:

Extended Data Fig.9b – what is this showing the expression of? There is no label

Response:

Extended Data Fig. 10b (previously Extended Data Fig. 9b) has now been explicitly labelled to indicate that it shows the fold change in *c-KIT* RNA expression across the *BRCA* survival groups (*BRCA1*-LTS/STS, *BRCA2*-LTS/STS, *BRCA*-P-LTS/STS), consistent with the description in the figure legend. The updated label clarifies the gene being assessed and the comparison being displayed.

Reviewer #1

Comment 1:

In their point by point response the authors have made all reasonable attempts to either clarify prior analyses and/or explain limitations of their study. With this in mind I have no further comments to make other than I feel their manuscript would make a useful addition to the literature.

Response:

We thank the reviewer for their positive and supportive assessment of our manuscript. We appreciate their time and thoughtful evaluation and are pleased that they consider our work to be a useful contribution to the literature.

Reviewer #2

Comment 1:

The authors did not respond to my Major comment 7: In Figure 5, it appears that activated dendritic cells are very high in IMMB.5 which has the worst survival but in Extended Data Figure 9, there does not appear to be an effect of activated dendritic cells on survival; however, it is stated in the text that mast cells are the cell type most strongly associated with short survival – it is curious as the heatmap does not seem to support this conclusion. The authors explained a response but did not edit the text. I suggest adding that the association with dendritic cells did not remain significant in the multivariable Cox model on page 24. The authors responded well to all other reviewer comments.

Response:

We thank the reviewer for raising this point. We have revised the Results section to explicitly clarify that, although activated dendritic cells are elevated in the poorest-survival immune cluster (IMMB.5), their association with survival does not remain significant in multivariable Cox regression analysis. Rather, resting mast cells emerge as the immune cell type most strongly and independently associated with short survival. This clarification has now been added to the main text on page 24, lines 562-566.